# EFFICIENT STOCHASTIC ALGORITHMS FOR CONTINUAL FINITE-SUM MINIMAX OPTIMIZATION

## ABSTRACT

This paper considers the continual finite-sum convex-concave minimax optimization. We seek a sequence $(\mathbf{x}_1^*, \mathbf{y}_1^*), \ldots, (\mathbf{x}_n^*, \mathbf{y}_n^*)$ which corresponds to the saddle points of prefix-sum functions $\{g_i(\mathbf{x}, \mathbf{y}) := \sum_{j=1}^i f_j(\mathbf{x}, \mathbf{y})/i\}_{i=1}^n$, where each component function $f_j: \mathbb{R}^{d_x} \times \mathbb{R}^{d_y} \to \mathbb{R}$ is strongly-convex-strongly-concave and feasible sets $\mathcal{X} \subseteq \mathbb{R}^{d_x}$ and $\mathcal{Y} \subseteq \mathbb{R}^{d_y}$ are convex and compact. We propose an efficient stochastic first-order algorithm that finds a sequence of $\epsilon$-saddle points for the continual finite-sum minimax optimization problem. In particular, our approach sparsely constructs the full gradient across all stages, and it leverages a novel extragradient iteration to achieve a sharper incremental first-order oracle complexity compared with existing methods. We also extend our methods to solve the continual finite-sum minimax optimization problem in the general convex-concave setting. Furthermore, we conduct numerical experiments that demonstrate the effectiveness of our approaches.

## 1 INTRODUCTION

In recent years, we have witnessed a surge of interest in solving the following finite-sum minimax optimization problem

$$\min_{\mathbf{x} \in \mathcal{X}} \max_{\mathbf{y} \in \mathcal{Y}} f(\mathbf{x}, \mathbf{y}) := \frac{1}{n} \sum_{i=1}^n f_i(\mathbf{x}, \mathbf{y}), \tag{1}$$

where both $\mathcal{X} \subseteq \mathbb{R}^{d_x}$ and $\mathcal{Y} \subseteq \mathbb{R}^{d_y}$ are convex and compact. This formulation is ubiquitous in various machine learning models, including AUC maximization (Guo et al., 2020; Hanley & McNeil, 1982; Ying et al., 2016), robust optimization (Duchi & Namkoong, 2019; Yan et al., 2019), adversarial learning (Sinha et al., 2017), and reinforcement learning (Sutton, 2018).

In many modern machine learning applications, new data keeps coming over time, and the objective function is dynamically evolving. The finite-sum objective function (1) fails to capture such a scenario since it requires all data to be accessible anytime. Therefore, we consider the continual finite-sum minimax optimization problem, which aims to build a continuously updated model that performs equally well on both new and past data (Castro et al., 2018; Rosenfeld & Tsotsos, 2018; Mavrothalassitis et al., 2024; Wang et al., 2024). Formally, we aim to solve the following prefix-sum minimax optimization problem

$$\min_{\mathbf{x} \in \mathcal{X}} \max_{\mathbf{y} \in \mathcal{Y}} g_i(\mathbf{x}, \mathbf{y}) := \frac{1}{i} \sum_{j=1}^i f_j(\mathbf{x}, \mathbf{y}) \tag{2}$$

at each stage $i \in [n]$, where each component function $f_j$ is $\mu$-strongly-convex-strongly-concave, $L$-smooth, and $G$-Lipschitz over the domain $\mathcal{X} \times \mathcal{Y}$. We denote the point $(\mathbf{x}_i^*, \mathbf{y}_i^*)$ to be the optimal solution of the prefix-sum minimax optimization problem (2) at stage $i$.

One straightforward solution for the continual finite-sum minimax optimization problem (2) is to apply existing stochastic first-order methods for finite-sum minimax problems on each prefix-sum problem. In particular, directly applying the SVRG/SAGA method (Palaniappan & Bach, 2016) to solve the problem (1) at each stage leads to a total incremental first-order oracle (IFO) complexity of $\mathcal{O}\big((n^2 + n^{3/2}\varkappa^2) \log(1/\epsilon)\big)$ where $\varkappa := L/\mu$ is the condition number. Similarly, we can apply

Table 1: We compare the incremental first-order oracle complexity for solving the continual finite-sum minimax optimization problem when each component function $f_i$ is $\mu$-strongly-convex-strongly-concave. We use the convergence metric $\mathbb{E}[||z_k - z^*||^2]$ for all methods.

| METHODS | IFO COMPLEXITY | REFERENCE |
|---|---|---|
| SVRG/SAGA | $\tilde{\mathcal{O}}\left(\left(n^2 + \dfrac{nL^2}{\mu^2}\right)\right)$ | PALANIAPPAN & BACH (2016) |
| A-SVRG/A-SAGA | $\tilde{\mathcal{O}}\left(\left(n^2 + \dfrac{n^{\frac{3}{2}}L}{\mu}\right)\right)$ | PALANIAPPAN & BACH (2016) |
| L-SVRE | $\tilde{\mathcal{O}}\left(\left(n^2 + \dfrac{n^{\frac{3}{2}}L}{\mu}\right)\right)$ | ALACAOGLU & MALITSKY (2022) |
| CSVRG | $\tilde{\mathcal{O}}\left(\dfrac{L^2 G}{\mu^{\frac{5}{2}}\epsilon^{\frac{1}{2}}} + \dfrac{nL^2}{\mu^2} + \dfrac{nG^{\frac{2}{3}}L^{\frac{2}{3}}}{\mu\epsilon^{\frac{1}{3}}}\right)$ | COROLLARY 5.4 |
| CSVRE | $\tilde{\mathcal{O}}\left(\dfrac{LG}{\mu^{\frac{3}{2}}\epsilon^{\frac{1}{2}}} + \dfrac{nL^2}{\mu^2} + \dfrac{nG^{\frac{2}{3}}L^{\frac{2}{3}}}{\mu\epsilon^{\frac{1}{3}}}\right)$ | COROLLARY 5.6 |

the loopless stochastic variance reduced extragradient (L-SVRE) (Alacaoglu & Malitsky, 2022) to achieve an improved complexity of $\mathcal{O}\big((n^2 + n^{3/2}\varkappa)\log(1/\epsilon)\big)$. In the continual learning setting where the number of functions $n$ is extremely large and $\epsilon$ is mediocre, the IFO complexities of both methods are prohibitively expensive.

In this work, we propose an efficient stochastic first-order method called continual stochastic variance reduced gradient (CSVRG) for the continual finite-sum minimax optimization problem where each component function $f_i(\mathbf{x})$ is strongly-convex-strongly-concave (SCSC). One of the major computational burdens of the aforementioned methods for the continual optimization problem is due to the evaluation of the full gradient at the snapshot variable during each stage, which leads to a $\mathcal{O}(n^2)$ factor in the IFO complexity. To alleviate this issue, we update the snapshot variable sparsely across all stages and construct the corresponding full gradient at these variables. We show that our method obtains a sequence of $\epsilon$-saddle points for the continual finite-sum minimax optimization problem with an IFO complexity of $\tilde{\mathcal{O}}(L^2 G\mu^{-\frac{5}{2}}\epsilon^{-\frac{1}{2}} + n\varkappa^2 + nG^{\frac{2}{3}}L^{\frac{2}{3}}\mu^{-1}\epsilon^{-\frac{1}{3}})$. Furthermore, we propose a new stochastic first-order algorithm called the continual stochastic variance-reduced extragradient (CSVRE) method, which extends CSVRG by incorporating a novel extragradient (EG) iteration. This step balances historical and current information in updating the auxiliary variable $z_i^{t+1/2}$, a key mechanism for achieving improved IFO complexity in the continual learning setting. The method computes a sequence of $\epsilon$-saddle points for Problem (2) with a reduced IFO complexity of $\tilde{\mathcal{O}}(LG\mu^{-\frac{3}{2}}\epsilon^{-\frac{1}{2}} + n\varkappa^2 + nG^{\frac{2}{3}}L^{\frac{2}{3}}\mu^{-1}\epsilon^{-\frac{1}{3}})$. It is worth noting that the dependency on $\mathcal{O}(n\epsilon^{-1/3})$ in our IFO complexity is close to the lower bound of $\mathcal{O}(n\epsilon^{-1/4})$ (Mavrothalassitis et al., 2024). More generally, if we only assume that each component function $f_i$ is convex in $\mathbf{x}$ and concave in $\mathbf{y}$, CSVRE could find a sequence of $\epsilon$-suboptimal solutions with $\tilde{\mathcal{O}}(L^{\frac{3}{2}}G(G^{\frac{1}{2}}+L^{\frac{1}{2}})\epsilon^{-3} + nG^{\frac{2}{3}}L(G^{\frac{1}{3}}+L^{\frac{1}{3}})\epsilon^{-2})$ IFO calls. We compare the proposed methods with existing algorithms for continual finite-sum minimax optimization in Table 1 and Table 2, which correspond to the strongly-convex–strongly-concave and general convex–concave settings, respectively. The results indicate that our methods attain strictly improved IFO complexity relative to the existing baselines when the number of stages $n$ is large and the target accuracy parameter $\epsilon$ is of moderate scale. Finally, we conduct numerical experiments on robust linear regression and fairness-aware machine learning problems to substantiate the practical effectiveness of the proposed approaches.

**Paper Organization**  In Section 2, we provide a literature review on continual learning and finite-sum minimax optimization. In Section 3, we formalize the notations and assumptions of our problem. In Section 4, we propose the CSVRG and CSVRE methods for solving the continual finite-sum minimax optimization problem. In Section 5, we provide the theoretical convergence analysis of both CSVRG and CSVRE methods. In Section 7, we conduct numerical experiments to validate the effectiveness of our approaches. Finally, we conclude our work in Section 8.

Table 2: We compare the incremental first-order oracle complexity for solving the continual finite-sum minimax optimization problem when each component function $f_i$ is convex-concave.

| METHODS | IFO COMPLEXITY | REFERENCE |
|---|---|---|
| L−SVRE | $\mathcal{O}\left(n^2 + \dfrac{n^{\frac{3}{2}}L}{\epsilon}\right)$ | ALACAOGLU & MALITSKY (2022) |
| CSVRG | $\tilde{\mathcal{O}}\left(\dfrac{L^{\frac{5}{2}}G(G^{\frac{1}{2}} + L^{\frac{1}{2}})}{\epsilon^4} + \dfrac{nG^{\frac{2}{3}}L(G^{\frac{1}{3}} + L^{\frac{1}{3}})}{\epsilon^2}\right)$ | COROLLARY 6.2 |
| CSVRE | $\tilde{\mathcal{O}}\left(\dfrac{L^{\frac{3}{2}}G(G^{\frac{1}{2}} + L^{\frac{1}{2}})}{\epsilon^3} + \dfrac{nG^{\frac{2}{3}}L(G^{\frac{1}{3}} + L^{\frac{1}{3}})}{\epsilon^2}\right)$ | COROLLARY 6.3 |

## 2 RELATED WORK

In this section, we review related work on continual learning and stochastic algorithms for finite-sum minimax optimization problems.

### 2.1 CONTINUAL/INCREMENTAL LEARNING

Although machine learning approaches have made significant progress in recent years, they are known to suffer from a phenomenon called catastrophic forgetting, a dramatic decrease in performance when new data or information is seen incrementally (Castro et al., 2018; Goodfellow et al., 2013; Kirkpatrick et al., 2017; McCloskey & Cohen, 1989; Mermillod et al., 2013). Several approaches (Jung et al., 2016; Li & Hoiem, 2017; Rusu et al., 2016; Terekhov et al., 2015; Kirkpatrick et al., 2017; Rebuffi et al., 2017) were introduced to mitigate this issue empirically without giving a theoretical guarantee. Recently, Mavrothalassitis et al. (2024) introduced efficient stochastic algorithms to address the finite-sum minimization problem in the continual learning setting, and established explicit bounds on the IFO complexity. While continual learning shares similarities with online learning in updating models sequentially, the two frameworks differ in their fundamental objectives. Continual learning prioritizes long-term knowledge retention and stable performance across a sequence of tasks or stages (Wang et al., 2024), seeking to prevent forgetting previously acquired knowledge. In contrast, online learning focuses on rapid adaptation to newly arriving data, often under adversarial or non-stationary conditions, with performance measured by cumulative regret (Hazan et al., 2016). These differences highlight the distinct challenges and evaluation criteria that characterize each framework.

### 2.2 FINITE-SUM MINIMAX OPTIMIZATION

Stochastic first-order algorithms for the finite-sum minimization problem are well-studied in the literature (Allen-Zhu, 2018; Defazio et al., 2014; Fang et al., 2018; Johnson & Zhang, 2013; Lin et al., 2018). It has been shown that solving the convex and nonconvex minimization problems with the stochastic recursive gradient estimator achieves the optimal IFO complexity (Allen-Zhu, 2018; Woodworth & Srebro, 2016; Fang et al., 2018; Wang et al., 2019; Arjevani et al., 2023). For finite-sum convex–concave minimax optimization, stochastic variance-reduced algorithms similarly achieve the best-known IFO complexities across various settings (Alacaoglu & Malitsky, 2022; Luo et al., 2020; 2021; Yang et al., 2020). Considering the $\mu$-strongly-convex-strongly-concave objective function, Palaniappan & Bach (2016) leveraged the variance reduction methodology for the minimax optimization problem (1) and they achieved an IFO complexity of $\mathcal{O}((n + \varkappa^2)\log(1/\epsilon))$. Furthermore, they applied the catalyst acceleration framework to their methods and obtained an improved IFO complexity of $\tilde{\mathcal{O}}((n + \varkappa\sqrt{n})\log(1/\epsilon))$. Later, the L-SVRE method (Alacaoglu & Malitsky, 2022) incorporated the extragradient iteration into the variance reduction framework and achieved an IFO complexity of $\mathcal{O}((n + \varkappa\sqrt{n})\log(1/\epsilon))$, which matches the lower bound for the finite-sum minimax optimization problems (Han et al., 2024). Recently, Chen & Luo (2024) proposed a stochastic algorithm called RAIN that attain a near-optimal first-order complexity of $\tilde{\mathcal{O}}(\sigma^2\epsilon^{-2})$ for computing $\epsilon$-stationary points, measured by the gradient norm of the objective function. Their

framework, however, is developed for the unconstrained setting, whereas our approach is designed for constrained problems. It is unclear whether their techniques can be extended to the constrained setting (Alacaoglu et al., 2024).

While progress has been made in the convex–concave case, the more general nonconvex–nonconcave minimax problem is intractable even in the unconstrained setting (Daskalakis et al., 2021). However, it is possible to introduce additional assumptions to solve the nonconvex-nonconcave minimax problem, such as the weak Minty variational inequality condition (Diakonikolas et al., 2021).

## 3 NOTATIONS AND ASSUMPTIONS

In this section, we formalize the notations and assumptions throughout this paper.

**Definition 3.1.** *For any differentiable function $\psi \colon \mathbb{R}^d \to \mathbb{R}$, we say $\psi$ is L-smooth for some $L > 0$ if for any $\mathbf{z}_1, \mathbf{z}_2 \in \mathbb{R}^d$, it holds that $\|\nabla\psi(\mathbf{z}_1) - \nabla\psi(\mathbf{z}_2)\| \leq L \|\mathbf{z}_1 - \mathbf{z}_2\|$.*

**Definition 3.2.** *For a differentiable function $\psi \colon \mathbb{R}^d \to \mathbb{R}$, we say $\psi$ is convex if for any $\mathbf{z}_1, \mathbf{z}_2 \in \mathbb{R}^d$, it holds that $\psi(\mathbf{z}_1) \geq \psi(\mathbf{z}_2) + \langle \nabla\psi(\mathbf{z}_2), \mathbf{z}_1 - \mathbf{z}_2 \rangle$. We say $\psi$ is $\mu$-strongly-convex for some $\mu > 0$ if $\psi(\cdot) - \frac{\mu}{2}\|\cdot\|^2$ is convex. We also say $\psi$ is concave ($\mu$-strongly-concave) if $-\psi$ is convex ($\mu$-strongly-convex).*

**Definition 3.3.** *For any function $f \colon \mathbb{R}^{d_x} \times \mathbb{R}^{d_y} \to \mathbb{R}$ and $\mu \geq 0$, we say $f$ is $\mu$-strongly-convex-strongly-concave if for any $\mathbf{x} \in \mathbb{R}^{d_x}$ and $\mathbf{y} \in \mathbb{R}^{d_y}$, it holds that $f(\mathbf{x}, \cdot)$ is $\mu$-strongly-concave and $f(\cdot, \mathbf{y})$ is $\mu$-strongly-convex.*

In the above definition, we allow $\mu$ to be zero. The notation 0-strongly-convex-strongly-concave then means general convex-concave. For the $\mu$-strongly-convex-strongly-concave setting with $\mu > 0$, each prefix-sum function has a unique saddle point, and our goal is to find a sequence of approximate saddle points sufficiently close to saddle points in terms of the weighted Euclidean distance.

**Definition 3.4.** *For the $\mu$-strongly-convex-strongly-concave minimax optimization problem $\min_{\mathbf{x} \in \mathcal{X}} \max_{\mathbf{y} \in \mathcal{Y}} f(\mathbf{x}, \mathbf{y})$ where $\mu > 0$, a point $(\hat{\mathbf{x}}, \hat{\mathbf{y}}) \in \mathcal{X} \times \mathcal{Y}$ is said to be an $\epsilon$-saddle point if it satisfies $\mu\|\hat{\mathbf{x}} - \mathbf{x}^*\|^2 + \mu\|\hat{\mathbf{y}} - \mathbf{y}^*\|^2 \leq \epsilon$, where $(\mathbf{x}^*, \mathbf{y}^*)$ is the optimal solution to the problem.*

For ease of presentation, we introduce the following notations.

**Definition 3.5.** *We let $\mathbf{z} = (\mathbf{x}, \mathbf{y}) \in \mathcal{X} \times \mathcal{Y}$ and define the gradient operator of the component $f_i$ as $h_i(\mathbf{z}) = [\nabla_{\mathbf{x}} f_i(\mathbf{x}, \mathbf{y})^\top, -\nabla_{\mathbf{y}} f_i(\mathbf{x}, \mathbf{y})^\top]^\top$.*

In the remainder of this paper, we always suppose Problem (2) satisfies the following assumptions.

**Assumption 3.6.** *The feasible sets $\mathcal{X}$ and $\mathcal{Y}$ in Problem (2) are convex and compact, and their diameters are bounded by $D_\mathcal{X}$ and $D_\mathcal{Y}$, respectively.*

**Assumption 3.7.** *For each $i \in [n]$, the function $f_i(\mathbf{x}, \mathbf{y})$ is L-smooth, $\mu$-strongly-convex-strongly-concave with $\mu \geq 0$, and G-Lipschitz continuous on $\mathcal{X} \times \mathcal{Y}$.*

## 4 METHODOLOGY

In this section, we propose stochastic first-order algorithms for solving the continual finite-sum strongly-convex-strongly-concave minimax optimization problem. We defer the extension to the convex-concave setting in Appendix D.

### 4.1 THE ALGORITHM

A simple approach to the continual optimization problem is to apply SVRG (Palaniappan & Bach, 2016) or L-SVRE (Alacaoglu & Malitsky, 2022) to the prefix-sum objective (2) at each stage $i$. At stage $i$, we maintain an aggregated snapshot variable $\tilde{\mathbf{z}}$ (updated every $\Theta(i)$ iterations) and compute the full prefix-sum gradient operator $\tilde{\nabla}_i = \frac{1}{i}\sum_{j=1}^{i} h_j(\tilde{\mathbf{z}})$. Using this snapshot, the L-SVRE method yields an $\epsilon$-approximate solution with $\mathcal{O}((i + \sqrt{i}\varkappa)\log(1/\epsilon))$ IFO calls at stage $i$. Summing over $i = 1$ to $n$ gives a total IFO complexity of $\mathcal{O}((n^2 + n^{\frac{3}{2}}\varkappa)\log(1/\epsilon))$, which is dominated by the $n^2$ term when $n$ is large and the required accuracy is moderate.

**Algorithm 1:** Continual Sparse Learning Method (CSL)

**1 Input**: $\hat{\mathbf{z}}_0 = (\hat{\mathbf{x}}_0, \hat{\mathbf{y}}_0) \in \mathcal{X} \times \mathcal{Y}$, prev $\leftarrow 0$, flag $\leftarrow$ false, sequences $\{T_i\}_{i=2}^n, \{\gamma_t\}$

**2** $\hat{\mathbf{z}}_1 \leftarrow \text{ExtraGradient}(\hat{\mathbf{z}}_0), \tilde{\nabla}_1 \leftarrow h_1(\hat{\mathbf{z}}_1)$

**3 for** $i = 2, \ldots, n$ **do**

**4**     **if** $i - \text{prev} \geq \alpha \cdot i$ **then**

**5**        $\tilde{\nabla}_{i-1} \leftarrow \frac{1}{i-1} \sum_{j=1}^{i-1} h_j(\hat{\mathbf{z}}_{i-1})$

**6**        prev $\leftarrow i - 1$

**7**        flag $\leftarrow$ true

**8**     **end**

**9**     **Option I**: $\hat{\mathbf{z}}_i \leftarrow \text{SVRG}(\hat{\mathbf{z}}_{\text{prev}}, \tilde{\nabla}_{i-1}, T_i, \{\gamma_t\}, \hat{\mathbf{z}}_{i-1})$

**10**     **Option II**: $\hat{\mathbf{z}}_i \leftarrow \text{SVRE}(\hat{\mathbf{z}}_{\text{prev}}, \tilde{\nabla}_{i-1}, T_i, \{\gamma_t\}, \hat{\mathbf{z}}_{i-1})$

**11**     **Output:** $\hat{\mathbf{z}}_i$.

**12**     **if** flag **then**

**13**        $\tilde{\nabla}_i \leftarrow \frac{1}{i} \sum_{j=1}^{i} h_j(\hat{\mathbf{z}}_i)$

**14**        prev $\leftarrow i$

**15**        flag $\leftarrow$ false

**16**     **end**

**17**     **else**

**18**        $\tilde{\nabla}_i \leftarrow (1 - \frac{1}{i})\tilde{\nabla}_{i-1} + \frac{1}{i}h_i(\hat{\mathbf{z}}_{\text{prev}})$

**19**     **end**

**20 end**

To reduce the expensive full-gradient operator evaluations required by L-SVRE, we extend the CSVRG method of Mavrothalassitis et al. (2024), originally developed for continual convex minimization, to our continual convex–concave minimax optimization. Following the standard CSVRG framework (Mavrothalassitis et al., 2024), we set a hyperparameter $\alpha > 0$ and update the aggregated snapshot variable, along with its full prefix-sum gradient operator

$$\tilde{\nabla}_{i-1} := \begin{pmatrix} \tilde{\nabla}_{x,i-1} \\ \tilde{\nabla}_{y,i-1} \end{pmatrix} = \begin{pmatrix} \frac{1}{i-1} \sum_{j=1}^{i-1} \nabla_{\mathbf{x}} f_j(\hat{\mathbf{x}}_{i-1}, \hat{\mathbf{y}}_{i-1}) \\ -\frac{1}{i-1} \sum_{j=1}^{i-1} \nabla_{\mathbf{y}} f_j(\hat{\mathbf{x}}_{i-1}, \hat{\mathbf{y}}_{i-1}) \end{pmatrix}, \tag{3}$$

whenever the condition $i - \text{prev} \geq \alpha i$ holds, where prev is the stage at which the snapshot was last updated. Using this full gradient operator (3), we construct inexpensive stochastic variance-reduced gradient estimators for both the convex function $g_i(\cdot, \mathbf{y})$ and the concave function $-g_i(\mathbf{x}, \cdot)$. In particular, the aggregated stochastic gradient estimators take the form

$$\nabla_i^t = \begin{pmatrix} \left(1 - \frac{1}{i}\right)\left(\nabla_{\mathbf{x}} f_{u_t}(\mathbf{x}_i^t, \mathbf{y}_i^t) - \nabla_{\mathbf{x}} f_{u_t}(\hat{\mathbf{x}}_{\text{prev}}, \hat{\mathbf{y}}_{\text{prev}}) + \tilde{\nabla}_{\mathbf{x},i-1}\right) + \frac{1}{i}\nabla_{\mathbf{x}} f_i(\mathbf{x}_i^t, \mathbf{y}_i^t) \\ \left(1 - \frac{1}{i}\right)\left(-\nabla_{\mathbf{y}} f_{u_t}(\mathbf{x}_i^t, \mathbf{y}_i^t) + \nabla_{\mathbf{y}} f_{u_t}(\hat{\mathbf{x}}_{\text{prev}}, \hat{\mathbf{y}}_{\text{prev}}) + \tilde{\nabla}_{\mathbf{y},i-1}\right) - \frac{1}{i}\nabla_{\mathbf{y}} f_i(\mathbf{x}_i^t, \mathbf{y}_i^t) \end{pmatrix} \tag{4}$$

where $u_t$ is drawn uniformly from $[i - 1]$. We then perform one projected stochastic gradient descent step on $\mathbf{x}_i^t$ and one projected stochastic gradient ascent step on $\mathbf{y}_i^t$ simultaneously to obtain the next iterate $\mathbf{z}_i^{t+1} = (\mathbf{x}_i^{t+1}, \mathbf{y}_i^{t+1})$. We refer to the whole procedure as the continual stochastic variance-reduced gradient (CSVRG) method, described in detail in Algorithm 1 under Option I.

## 4.2 ACCELERATION WITH EXTRAGRADIENT

In this subsection, we introduce a novel extragradient iteration into the continual learning framework (Algorithm 1) to obtain a more efficient stochastic algorithm. The main intuition of the stochastic variance reduced extragradient is to use the gradient at the snapshot variable to find a mid-point, and then use the gradient at the mid-point to find the next iterate. In particular, we modify the subroutine SVRG method (Algorithm 2) by introducing two auxiliary variables $\bar{\mathbf{z}}_i^t$ and $\mathbf{z}_i^{t+1/2}$ at each iteration. The variable $\bar{\mathbf{z}}_i^t$ is defined as the weighted average of $\mathbf{z}_i^t$ and $\hat{\mathbf{z}}_{\text{prev}}$, given by $\bar{\mathbf{z}}_i^t = \eta_t \mathbf{z}_i^t + (1 - \eta_t)\hat{\mathbf{z}}_{\text{prev}}$,

---

**Algorithm 2:** $\mathrm{SVRG}(\hat{\mathbf{z}}_{\mathrm{prev}}, \tilde{\nabla}_{i-1}, T_i, \{\gamma_t\}, \hat{\mathbf{z}}_{i-1})$

---

**1** $\mathbf{z}_i^0 \leftarrow \hat{\mathbf{z}}_{i-1}$

**2 for** $t = 0, \dots, T_i - 1$ **do**

**3**      Select $u_t \sim \mathrm{Unif}(1, \dots, i-1)$

**4**      $\nabla_i^t \leftarrow \left(1 - \frac{1}{i}\right)\left(h_{u_t}(\mathbf{z}_i^t) - h_{u_t}(\hat{\mathbf{z}}_{\mathrm{prev}}) + \tilde{\nabla}_{i-1}\right) + \frac{1}{i}h_i(\mathbf{z}_i^t)$

**5**      $\mathbf{z}_i^{t+1} \leftarrow \Pi_{\mathcal{X} \times \mathcal{Y}}(\mathbf{z}_i^t - \gamma_t \nabla_i^t)$

**6 end**

**7 return** $\hat{\mathbf{z}}_i \leftarrow \mathbf{z}_i^{T_i}$.

---

**Algorithm 3:** $\mathrm{SVRE}(\hat{\mathbf{z}}_{\mathrm{prev}}, \tilde{\nabla}_{i-1}, T_i, \{\gamma_t\}, \hat{\mathbf{z}}_{i-1})$

---

**1** $\mathbf{z}_i^0 \leftarrow \hat{\mathbf{z}}_{i-1}$

**2 for** $t = 0, \dots, T_i - 1$ **do**

**3**      $\bar{\mathbf{z}}_i^t \leftarrow \eta_t \mathbf{z}_i^t + (1 - \eta_t)\hat{\mathbf{z}}_{\mathrm{prev}}$

**4**      $\mathbf{z}_i^{t+1/2} \leftarrow \Pi_{\mathcal{X} \times \mathcal{Y}}\left(\bar{\mathbf{z}}_i^t - \gamma_t\left(\left(1 - \frac{1}{i}\right)\tilde{\nabla}_{i-1} + \frac{1}{i}h_i(\hat{\mathbf{z}}_{\mathrm{prev}})\right)\right)$

**5**      Select $u_t \sim \mathrm{Unif}(1, \dots, i-1)$

**6**      $\nabla_i^t \leftarrow \left(1 - \frac{1}{i}\right)\left(h_{u_t}\left(\mathbf{z}_i^{t+1/2}\right) - h_{u_t}(\hat{\mathbf{z}}_{\mathrm{prev}}) + \tilde{\nabla}_{i-1}\right) + \frac{1}{i}h_i\left(\mathbf{z}_i^{t+1/2}\right)$

**7**      $\mathbf{z}_i^{t+1} \leftarrow \Pi_{\mathcal{X} \times \mathcal{Y}}(\bar{\mathbf{z}}_i^t - \gamma_t \nabla_i^t)$

**8 end**

**9 return** $\hat{\mathbf{z}}_i \leftarrow \mathbf{z}_i^{T_i}$.

---

where $\eta_t \in [0, 1)$ is a weighting parameter. Then we find the mid-point $\mathbf{z}_i^{t+1/2}$ with the following novel extragradient iteration:

$$\mathbf{z}_i^{t+1/2} = \Pi_{\mathcal{X} \times \mathcal{Y}}\left(\bar{\mathbf{z}}_i^t - \gamma_t\left(\left(1 - \frac{1}{i}\right)\tilde{\nabla}_{i-1} + \frac{1}{i}h_i(\hat{\mathbf{z}}_{\mathrm{prev}})\right)\right). \tag{5}$$

We remark that $\sum_{j=1}^i h_j(\hat{\mathbf{z}}_{\mathrm{prev}})/i = (1 - 1/i)\tilde{\nabla}_{i-1} + h_i(\hat{\mathbf{z}}_{\mathrm{prev}})/i$ represents the full gradient of the prefix-sum objective function $g_i(\hat{\mathbf{z}}_{\mathrm{prev}})$. Recall that the classical extragradient method computes the intermediate point $\mathbf{z}_i^{t+1/2}$ solely based on the aggregated gradient $\tilde{\nabla}_{i-1}$. By contrast, update (5) incorporates a carefully weighted combination of both the historical information $\tilde{\nabla}_{i-1}$ and the gradient of $f_i(\hat{\mathbf{z}}_{\mathrm{prev}})$ at current stage. This refinement is crucial for obtaining an improved IFO complexity bound in the continual learning setting. Building upon this idea, we then construct the stochastic variance-reduced gradient estimator at the midpoint $\mathbf{z}_i^{t+1/2}$ in a manner analogous to formula (4), such that

$$\nabla_i^t = \left(1 - \frac{1}{i}\right)\left(h_{u_t}\left(\mathbf{z}_i^{t+1/2}\right) - h_{u_t}(\hat{\mathbf{z}}_{\mathrm{prev}}) + \tilde{\nabla}_{i-1}\right) + \frac{1}{i}h_i\left(\mathbf{z}_i^{t+1/2}\right). \tag{6}$$

Finally, we perform one step of GDA to obtain the next iterate $\mathbf{z}_i^{t+1}$. We call the whole procedure continual stochastic variance-reduced extragradient (CSVRE) and present its detail in Algorithm 1 with Option II. In the next section, we will show that the CSVRE method obtains an improved IFO complexity compared with the CSVRG method.

## 5 THEORETICAL ANALYSIS

In this section, we establish the IFO complexity of the proposed methods in Section 4 for solving the continual finite-sum minimax optimization problem where each component function $f_i$ is SCSC. We start our analysis with the following two lemmas, which show that the stochastic variance-reduced gradient estimators defined in (4) and (6) are unbiased and have controlled variance.

**Lemma 5.1.** *Let $u_t$ be some index uniformly sampled from $[i-1]$. Then for any $\mathbf{z} \in \mathcal{X} \times \mathcal{Y}$, the gradient estimator*

$$\bar{\nabla} = \left(1 - \frac{1}{i}\right)\left(h_{u_t}(\mathbf{z}) - h_{u_t}(\hat{\mathbf{z}}_{\text{prev}}) + \tilde{\nabla}_{i-1}\right) + \frac{1}{i}h_i(\mathbf{z})$$

*that satisfies the property $\mathbb{E}\left[\bar{\nabla}\right] = \sum_{j=1}^{i} h_j(\mathbf{z})/i$.*

The variance of the stochastic gradient estimator is bounded as follows.

**Lemma 5.2.** *For any $\mathbf{z} \in \mathcal{X} \times \mathcal{Y}$, then the gradient estimator $\bar{\nabla}$ defined in Lemma 5.1 satisfies*

$$\mathbb{E}\left[\left\|\bar{\nabla} - \frac{1}{i}\sum_{j=1}^{i} h_j(\mathbf{z})\right\|^2\right] \leq 2L^2\mathbb{E}\left[\|\mathbf{z} - \mathbf{z}_i^*\|^2\right] + \frac{32G^2L^2\alpha^2}{\mu^2} + 4L^2\mathbb{E}\left[\|\hat{\mathbf{z}}_{\text{prev}} - \mathbf{z}_{\text{prev}}^*\|^2\right],$$

*where $\alpha$ is some positive hyperparameter, $\mathbf{z}_i^*$ and $\mathbf{z}_{\text{prev}}^*$ are the saddle points at stages $i$ and $\text{prev}$, respectively.*

Lemma 5.2 implies that the variance of the stochastic gradient estimator decreases when the variable $\mathbf{z}$ is close to the optimum $\mathbf{z}_i^*$. Based on this result, we can perform the convergence analysis for each stage and obtain a sequence of $\epsilon$-saddle points under appropriate choices of hyperparameters for the CSVRG method.

**Theorem 5.3.** *Under Assumption 3.6 and Assumption 3.7 with $\mu > 0$, running the CSVRG method (Algorithm 1 with Option I) with $\alpha = \mu\epsilon^{\frac{1}{3}}G^{-\frac{2}{3}}L^{-\frac{2}{3}}$ and the subroutine SVRG method (Algorithm 2) with*

$$\gamma_t = \frac{2}{\mu(t+\beta)}, \ \beta = \frac{6L^2}{\mu^2}, \ T_i = \mathcal{O}\left(\frac{L^2G}{\mu^{\frac{5}{2}}i\epsilon^{\frac{1}{2}}} + \frac{L^2}{\mu^2} + \frac{G^{\frac{2}{3}}L^{\frac{2}{3}}}{\mu\epsilon^{\frac{1}{3}}}\right),$$

*then the output $\hat{\mathbf{z}}_i$ satisfies $\mathbb{E}\left[\mu\|\hat{\mathbf{z}}_i - \mathbf{z}_i^*\|^2\right] \leq \epsilon$ for each $i \in [n]$.*

Theorem 5.3 implies the following IFO complexity of Algorithm 1 with Option I.

**Corollary 5.4.** *Under Assumption 3.6 and Assumption 3.7 with $\mu > 0$, the CSVRG method (Algorithm 1 with Option I) requires at most*

$$\mathcal{O}\left(\frac{L^2G\log n}{\mu^{\frac{5}{2}}\epsilon^{\frac{1}{2}}} + \frac{nL^2}{\mu^2} + \frac{nG^{\frac{2}{3}}L^{\frac{2}{3}}\log n}{\mu\epsilon^{\frac{1}{3}}}\right)$$

*IFO calls to obtain a sequence of $\epsilon$-saddle points for solving the Problem (2).*

The extragradient method is known to achieve improved convergence rates for finite-sum strongly-convex-strongly-concave minimax optimization. A natural question is whether our novel extragradient step, inspired by the classical variant, can further reduce the IFO complexity in the continual learning setting. Specifically, we show that the CSVRE method attains a sequence of $\epsilon$-saddle points under the following hyperparameter choices.

**Theorem 5.5.** *Under Assumption 3.6 and Assumption 3.7 with $\mu > 0$, running the CSVRE method (Algorithm 1 with Option II) with $\alpha = \mu\epsilon^{\frac{1}{3}}G^{-\frac{2}{3}}L^{-\frac{2}{3}}$ and the subroutine SVRE method (Algorithm 3) with*

$$\gamma_t = \frac{2}{\mu(t+\beta)}, \quad \eta_t = 1 - 2\gamma_t^2 L^2, \quad \beta = \frac{8L}{\mu}, \quad T_i = \mathcal{O}\left(\frac{LG}{\mu^{\frac{3}{2}}i\epsilon^{\frac{1}{2}}} + \frac{L^2}{\mu^2} + \frac{G^{\frac{2}{3}}L^{\frac{2}{3}}}{\mu\epsilon^{\frac{1}{3}}}\right),$$

*then the output $\hat{\mathbf{z}}_i$ satisfies $\mathbb{E}\left[\mu\|\hat{\mathbf{z}}_i - \mathbf{z}_i^*\|^2\right] \leq \epsilon$ for each $i \in [n]$.*

Theorem 5.5 implies the following IFO complexity of Algorithm 1 with Option II.

**Corollary 5.6.** *Under Assumption 3.6 and Assumption 3.7 with $\mu > 0$, the CSVRE method (Algorithm 1 with Option II) requires at most*

$$\mathcal{O}\left(\frac{LG\log n}{\mu^{\frac{3}{2}}\epsilon^{\frac{1}{2}}} + \frac{nL^2}{\mu^2} + \frac{nG^{\frac{2}{3}}L^{\frac{2}{3}}\log n}{\mu\epsilon^{\frac{1}{3}}}\right)$$

*IFO calls to obtain a sequence of $\epsilon$-saddle points for solving the Problem (2).*

We now consider the lower bound for solving our minimax problem by using IFO algorithms. Specifially, we focus on the continual finite-sum minimax optimization problem in which the objective function at the $i$-th stage takes the form

$$f_i(x,y) = g_i(x) + h_i(y),$$

where $g_i(x)$ is $\mu$-strongly convex and $h_i(y)$ is $\mu$-strongly concave, which leads to minimizing $x$ and maximizing $y$ are independent. Applying Theorems 3 and 5 of (Mavrothalassitis et al., 2024), we know that finding an $\epsilon$-suboptimal solution of the continual minimization problem with the prefix functions $\{g_i(x)\}_{i=1}^n$ or $\{-h_i(x)\}_{i=1}^n$ requires at least $\Omega(n\epsilon^{-1/4})$ IFO calls. This implies finding $\epsilon$-stationary point of the continual minimax problem with prefix functions $\{f_i(x)\}_{i=1}^n$ also requires at least $\Omega(n\epsilon^{-1/4})$ IFO calls. Recall that Colloary 5.6 says the IFO complexity of our CSVRE method is dominated by the factor of $\mathcal{O}(n\epsilon^{-1/3})$, which is close to the lower bound of $\Omega(n\epsilon^{-1/4})$. However, how to fill the gap of $\epsilon^{-1/12}$ remains an open problem.

Recall that directly applying L-SVRE achieves an IFO complexity of $\mathcal{O}(n^2 \log(1/\epsilon))$. On the other hand, Theorems 3 of Mavrothalassitis et al. (2024) implies that for any $\alpha > 0$, there is no IFO method for the continual optimization problem can achieve the IFO complexity that is better than $\mathcal{O}(n^{2-\alpha} \log(1/\epsilon))$. Hence, the trade-off between the dependency on $n$ and $\epsilon$ can not be avoided. To improves the dependence on $n$, our proposed CSVRE method attains an IFO complexity of $\mathcal{O}(n\epsilon^{-1/3})$, while the results in Theorem 3 of Mavrothalassitis et al. (2024) means it is impossible to retain a logarithmic dependence on $1/\epsilon$ while still achieving a linear dependence on $n$.

# 6 EXTENSION TO THE GENERAL CONVEX-CONCAVE SETTING

This section extends the proposed stochastic algorithms in Section 4 to solve the continual finite-sum minimax optimization problem where each component function $f_i$ is convex in $\mathbf{x}$ and concave in $\mathbf{y}$. For the general convex-concave setting, we are interested in finding an approximate suboptimal solution in terms of the duality gap, which is defined as follows.

**Definition 6.1.** *For the convex-concave minimax optimization problem* $\min_{\mathbf{x} \in \mathcal{X}} \max_{\mathbf{y} \in \mathcal{Y}} f(\mathbf{x}, \mathbf{y})$, *the point* $(\hat{\mathbf{x}}, \hat{\mathbf{y}}) \in \mathcal{X} \times \mathcal{Y}$ *is said to be an $\epsilon$-suboptimal solution w.r.t. the duality gap if*

$$\max_{\mathbf{y} \in \mathcal{Y}} f(\hat{\mathbf{x}}, \mathbf{y}) - \min_{\mathbf{x} \in \mathcal{X}} f(\mathbf{x}, \hat{\mathbf{y}}) \leq \epsilon.$$

We formally present the IFO complexity of CSVRG for achieving an $\epsilon$-suboptimal solution in continual convex–concave minimax optimization as follows.

**Corollary 6.2.** *Under Assumption 3.6 and Assumption 3.7 with $\mu = 0$, the CSVRG method (Algorithm 4 with Option I) requires at most*

$$\mathcal{O}\left(\frac{L^{\frac{5}{2}} G(G^{\frac{1}{2}} + L^{\frac{1}{2}}) \log n}{\epsilon^4} + \frac{n G^{\frac{2}{3}} L(G^{\frac{1}{3}} + L^{\frac{1}{3}}) \log n}{\epsilon^2}\right)$$

*IFO calls to obtain a sequence of $\epsilon$-suboptimal solutions for solving the Problem (2).*

Similar to the SCSC setting, applying the CSVRE method to solve the continual finite-sum minimax optimization problem in the convex-concave setting leads to an improved IFO complexity. We formally present the IFO complexity of CSVRE for achieving an $\epsilon$-suboptimal solution in continual convex–concave minimax optimization as follows.

**Corollary 6.3.** *Under Assumption 3.6 and Assumption 3.7 with $\mu = 0$, the CSVRE method (Algorithm 4 with Option II) requires at most*

$$\mathcal{O}\left(\frac{L^{\frac{3}{2}} G(G^{\frac{1}{2}} + L^{\frac{1}{2}}) \log n}{\epsilon^3} + \frac{n G^{\frac{2}{3}} L(G^{\frac{1}{3}} + L^{\frac{1}{3}}) \log n}{\epsilon^2}\right)$$

*IFO calls to obtain a sequence of $\epsilon$-suboptimal solutions for solving the Problem (2).*

We remark that our method is better suited for the continual learning setting where $n$ is large and $\epsilon$ is moderate.

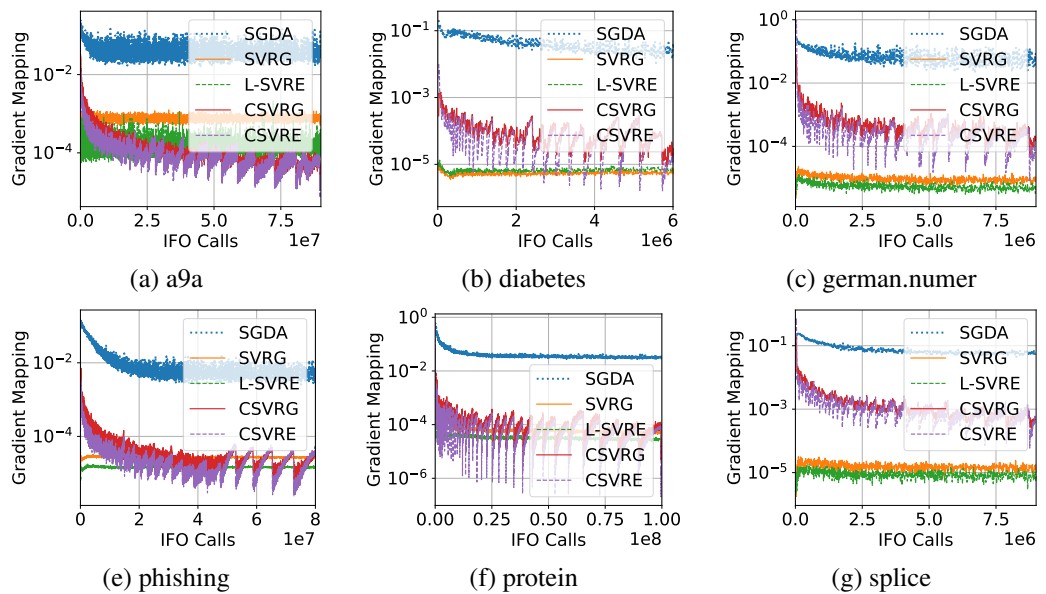

Figure 1: Gradient mapping vs the number of IFO calls for the robust linear regression problem.

## 7 EXPERIMENTS

In this section, we conduct numerical experiments to validate the effectiveness of the proposed methods. We compare our methods CSVRG and CSVRE with baseline methods including SGDA (Nemirovski, 2004; Korpelevich, 1976), SVRG (Palaniappan & Bach, 2016), and L-SVRE (Alacaoglu & Malitsky, 2022). We test all the methods on the problems of robust linear regression and fairness-aware machine learning.

### 7.1 ROBUST LINEAR REGRESSION

We consider the prefix-sum robust linear regression problem

$$\min_{\|\mathbf{x}\| \le R_x} \max_{\|\mathbf{y}\| \le R_y} g_i(\mathbf{x}, \mathbf{y}) := \frac{1}{i} \sum_{j=1}^{i} f_j(\mathbf{x}, \mathbf{y}),$$

where each component function is defined as

$$f_i(\mathbf{x}, \mathbf{y}) = \frac{1}{2}(\mathbf{x}^\top (\mathbf{a}_j + \mathbf{y}) - b_j)^2 + \lambda \|\mathbf{x}\|^2 - \beta \|\mathbf{y}\|^2.$$

The random variable $\mathbf{x} \in \mathbb{R}^d$ is the model weight, $\mathbf{y} \in \mathbb{R}^d$ describes the noise, $\mathbf{a}_j \in \mathbb{R}^d$ is the data feature, and $b_j \in \mathbb{R}$ is the corresponding label. We test the algorithm on six real-world datasets ("a9a", "diabetes", "german.numer", "phishing", "protein", "splice") from the LIBSVM repository (Chang & Lin, 2011). We set hyperparameters $\lambda = 2.0$, $\beta = 2.0$, $R_x = 1.0$ and $R_y = 0.1$ across all the datasets. For both CSVRG and CSVRE methods, we set the sparsity parameter $\alpha = 0.1$. We choose $T_i = 4000$ for the CSVRG method and $T_i = 3000$ for the CSVRE method so that the per-stage IFO calls of the two methods match. At each stage, we choose the outer iteration to be 10 and the inner iterations to be 200. The step size for each algorithm is tuned from the set $\{1e-5, 3e-5, \ldots, 0.03, 0.1\}$ for each algorithm. Following the setup of Mavrothalassitis et al. (2024), we reveal a new data point at each stage $i \in [n]$. The experimental results are presented in Figure 1. Our methods may underperform relative to the baselines on datasets with small sample sizes (e.g., "diabete" with $n = 750$, "german.numer" with $n = 1,000$, "splice" with $n = 1,000$). However, they demonstrate clear and consistent superiority on larger datasets (e.g., "a9a" with $n = 13,000$, "phishing" with $n = 12,000$, "protein" with $n = 12,000$ ). This empirical trend aligns well with our theoretical results.

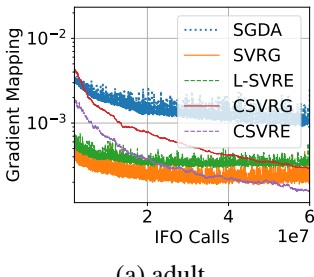 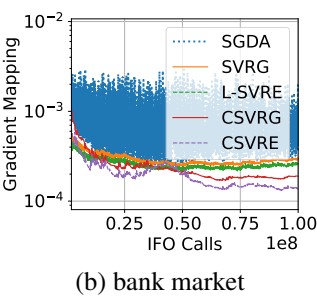 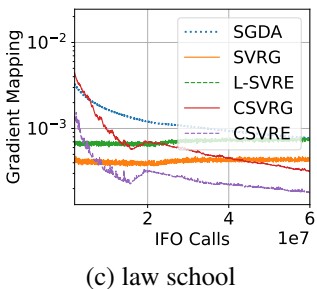

|  (a) adult | (b) bank market | (c) law school |

Figure 2: Gradient mapping vs the number of IFO calls for the fairness-aware machine learning problem.

### 7.2 FAIRNESS-AWARE MACHINE LEARNING

We consider the following prefix-sum fairness-aware machine learning problem:

$$\min_{\mathbf{x}\in\mathbb{R}^d} \max_{y\in\mathbb{R}} g_i(\mathbf{x}, y) \coloneqq \frac{1}{i} \sum_{j=1}^{i} f_j(\mathbf{x}, y),$$

where each component function is

$$f_j(\mathbf{x}, y) = \left( l(\mathbf{a}_j, b_j, \mathbf{x}) - \beta l(\mathbf{a}_j^\top \mathbf{x}, c_j, y) \right) + \lambda \left\| \mathbf{x} \right\|^2 - \gamma y^2,$$

and the loss $l$ is the logit functions: $l(\mathbf{a}, b, \mathbf{c}) = \log(1 + \exp(-b\mathbf{a}^\top\mathbf{c}))$. The tuple $(\mathbf{a}_j, b_j, c_j)$ is a training data point where $\mathbf{a}_j \in \mathbb{R}^d$ is the input variable, $b_j \in \mathbb{R}$ is the output, and $c_j \in \mathbb{R}$ is the input variable that we want to protect and make it unbiased. Our experiments are conducted on the fairness-aware binary classification datasets "adult", "bank market", and "law school" (Le Quy et al., 2022; Liu & Luo, 2022).We set the parameters $\beta$, $\lambda$ and $\gamma$ as 0.5, $10^{-4}$ and $10^{-4}$, respectively. For other hyperparameters, including stepsize and the number of iterations $T_i$, we use the same parameter setting in the robust linear regression experiment. The experimental results for this task are presented in Figure 2. As the number of IFO call increases, both the CSVRG and CSVRE methods demonstrate consistently superior performance compared with the baseline approaches, highlighting their effectiveness in leveraging larger datasets.

## 8 CONCLUSION

In this paper, we propose two efficient stochastic first-order algorithms, CSVRG and CSVRE, for continual finite-sum minimax optimization. Our theoretical analysis shows that these methods attain a sequence of approximate solutions with significantly fewer IFO calls than existing approaches in the strongly-convex-strongly-concave setting. Moreover, we extend the applicability of our framework to the more general convex-concave setting, broadening its relevance to a wider class of minimax problems.

In future work, it would be valuable to investigate lower bounds for stochastic algorithms in continual finite-sum minimax optimization. It would also be of interest to develop efficient stochastic algorithms for continual nonconvex-concave optimization problems.

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

APPENDIX

The appendix is organized as below. Section A introduces several key lemmas essential for the convergence analysis of the proposed methods for solving the continual minimax optimization problem. Section B provides the convergence analysis and total time complexity of the CSVRG method for solving the continual problem in the strongly-convex-strongly-concave setting. Section C provides the convergence analysis of the CSVRE method and demonstrates its improved time complexity by incorporating the extragradient iteration. Section D presents the details of the extensions of the proposed methods to the general convex-concave setting. We present some additional numerical experiments in Section E.

## A    ESTABLISHED RESULTS

Firstly, we present some useful tools for the analysis of the constrained optimization.

**Lemma A.1** (Nesterov et al. (2018)). *Given a convex and compact set $\mathcal{C} \subseteq \mathbb{R}^d$, and any $\mathbf{u}, \mathbf{v} \in \mathbb{R}^d$, we have*

$$\|\Pi_{\mathcal{C}}(\mathbf{u}) - \Pi_{\mathcal{C}}(\mathbf{v})\| \leq \|\mathbf{u} - \mathbf{v}\|. \tag{7}$$

**Lemma A.2** (Nesterov et al. (2018)). *Given a convex and compact set $\mathcal{C} \subseteq \mathbb{R}^d$, for any $\mathbf{u} \in \mathbb{R}^d$ and $\mathbf{v} \in \mathcal{C}$, we have*

$$\langle \Pi_{\mathcal{C}}(\mathbf{u}) - \mathbf{u}, \Pi_{\mathcal{C}}(\mathbf{u}) - \mathbf{v} \rangle \leq 0. \tag{8}$$

Next we provide some properties for convex-concave functions.

**Lemma A.3** (Lin et al. (2020)). *Assume that $f(\mathbf{x}, \mathbf{y}) \colon \mathcal{X} \times \mathcal{Y} \to \mathbb{R}$ is $L$-smooth and $\mu$-strongly-convex-strongly-concave. We define*

$$\mathbf{y}_f^*(\cdot) := \arg\max_{y \in \mathcal{Y}} f(\cdot, \mathbf{y}), \Phi_f(\cdot) := \max_{\mathbf{y} \in \mathcal{Y}} f(\cdot, \mathbf{y}),$$

$$\mathbf{x}_f^*(\cdot) := \arg\min_{\mathbf{x} \in \mathcal{X}} f(\mathbf{x}, \cdot), \Psi_f(\cdot) := \min_{\mathbf{x} \in \mathcal{X}} f(\mathbf{x}, \cdot).$$

*Then it holds that*

- *the function $\mathbf{y}_f^*(\cdot)$ is $\varkappa$-Lipschitz,*

- *the function $\Phi_f(\cdot)$ is $2\varkappa L$-smooth and $\mu$-strongly convex with $\nabla \Phi_f(\cdot) = \nabla_{\mathbf{x}} f(\cdot, \mathbf{y}_f^*(\cdot))$,*

- *the function $\mathbf{x}_f^*(\cdot)$ is $\varkappa$-Lipschitz,*

- *the function $\Psi_f(\cdot)$ is $2\varkappa L$-smooth and $\mu$-strongly concave with $\nabla \Psi_f(\cdot) = \nabla_{\mathbf{x}} f(\mathbf{x}_f^*(\cdot), \cdot)$.*

**Lemma A.4** (Rockafellar (1970)). *Under Assumption 3.7, the gradient operator $h_i(\cdot) = [\nabla_{\mathbf{x}} f_i(\cdot) \quad -\nabla_{\mathbf{y}} f_i(\cdot)]^{\top}$ holds that*

$$\langle h_i(\mathbf{z}_1) - h_i(\mathbf{z}_2), \mathbf{z}_1 - \mathbf{z}_2 \rangle \geq \mu \|\mathbf{z}_1 - \mathbf{z}_2\|^2$$

*for any $\mathbf{z}_1, \mathbf{z}_2 \in \mathcal{X} \times \mathcal{Y}$.*

## B    CONVERGENCE ANALYSIS OF CSVRG FOR SCSC CASE

In this section, we present the convergence analysis of the CSVRG method for solving the continual minimax optimization problem when each component function $f_j(\mathbf{x}, \mathbf{y})$ is strongly-convex-strongly-concave.

### B.1 Supporting Lemmas

Firstly, we show that the optimal solutions of prefix-sum objective functions $g_i(\mathbf{x}, \mathbf{y})$ and $g_j(\mathbf{x}, \mathbf{y})$ are similar when $i$ and $j$ are close to each other.

**Lemma B.1.** *For all $i \in [n-1]$ and $j \in [n-i]$, it holds that*

$$\sqrt{\mu \left\| \mathbf{z}_{i+j}^* - \mathbf{z}_i^* \right\|^2} \leq \frac{2\sqrt{2}jG}{\sqrt{\mu}(2i+j)}.$$

*Proof.* By the strong convexity of $g_{i+j}(\cdot, \mathbf{y})$ for any $\mathbf{y} \in \mathcal{Y}$ in Assumption 3.7, we have

$$\frac{\mu}{2} \left\| \mathbf{x}_{i+j}^* - \mathbf{x}_i^* \right\|^2 \leq g_{i+j}(\mathbf{x}_i^*, \mathbf{y}_{i+j}^*) - g_{i+j}(\mathbf{x}_{i+j}^*, \mathbf{y}_{i+j}^*) - \langle \nabla g_{i+j}(\mathbf{x}_{i+j}^*, \mathbf{y}_{i+j}^*), \mathbf{x}_i^* - \mathbf{x}_{i+j}^* \rangle$$
$$\leq g_{i+j}(\mathbf{x}_i^*, \mathbf{y}_{i+j}^*) - g_{i+j}(\mathbf{x}_{i+j}^*, \mathbf{y}_{i+j}^*),$$

the last inequality is due to the optimal choice of $\mathbf{x}_{i+j}^*$. Similarly, the strong concavity of $g_{i+j}(\mathbf{x}, \cdot)$ for any $\mathbf{x} \in \mathcal{X}$ implies that

$$\frac{\mu}{2} \left\| \mathbf{y}_{i+j}^* - \mathbf{y}_i^* \right\|^2 \leq - g_{i+j}(\mathbf{x}_{i+j}^*, \mathbf{y}_i^*) + g_{i+j}(\mathbf{x}_{i+j}^*, \mathbf{y}_{i+j}^*) + \langle \nabla g_{i+j}(\mathbf{x}_{i+j}^*, \mathbf{y}_{i+j}^*), \mathbf{y}_i^* - \mathbf{y}_{i+j}^* \rangle$$
$$\leq - g_{i+j}(\mathbf{x}_{i+j}^*, \mathbf{y}_i^*) + g_{i+j}(\mathbf{x}_{i+j}^*, \mathbf{y}_{i+j}^*).$$

Adding up the above two inequalities, we have

$$\frac{\mu}{2} \left\| \mathbf{x}_{i+j}^* - \mathbf{x}_i^* \right\|^2 + \frac{\mu}{2} \left\| \mathbf{y}_{i+j}^* - \mathbf{y}_i^* \right\|^2$$
$$\leq g_{i+j}(\mathbf{x}_i^*, \mathbf{y}_{i+j}^*) - g_{i+j}(\mathbf{x}_{i+j}^*, \mathbf{y}_i^*)$$
$$= \frac{1}{i+j} \sum_{k=1}^{i+j} f_k(\mathbf{x}_i^*, \mathbf{y}_{i+j}^*) - \frac{1}{i+j} \sum_{k=1}^{i+j} f_k(\mathbf{x}_{i+j}^*, \mathbf{y}_i^*)$$
$$= \frac{i}{i+j} \left( g_i(\mathbf{x}_i^*, \mathbf{y}_{i+j}^*) - g_i(\mathbf{x}_{i+j}^*, \mathbf{y}_i^*) \right) + \frac{1}{i+j} \sum_{k=i+1}^{i+j} \left( f_k(\mathbf{x}_i^*, \mathbf{y}_{i+j}^*) - f_k(\mathbf{x}_{i+j}^*, \mathbf{y}_i^*) \right)$$
$$= \frac{i}{i+j} \left( g_i(\mathbf{x}_i^*, \mathbf{y}_{i+j}^*) - g_i(\mathbf{x}_i^*, \mathbf{y}_i^*) \right) + \frac{i}{i+j} (g_i(\mathbf{x}_i^*, \mathbf{y}_i^*) - g_i(\mathbf{x}_{i+j}^*, \mathbf{y}_i^*)) + \frac{1}{i+j} \sum_{k=i+1}^{i+j} \left( f_k(\mathbf{x}_i^*, \mathbf{y}_{i+j}^*) - f_k(\mathbf{x}_{i+j}^*, \mathbf{y}_i^*) \right)$$
$$\leq \frac{i}{i+j} \left( -\frac{\mu}{2} \left\| \mathbf{y}_{i+j}^* - \mathbf{y}_i^* \right\|^2 + \langle \nabla_{\mathbf{y}} g_i(\mathbf{x}_i^*, \mathbf{y}_i^*), \mathbf{y}_{i+j}^* - \mathbf{y}_i^* \rangle \right) + \frac{i}{i+j} \left( -\frac{\mu}{2} \left\| \mathbf{x}_{i+j}^* - \mathbf{x}_i^* \right\|^2 - \langle \nabla_{\mathbf{x}} g_i(\mathbf{x}_i^*, \mathbf{y}_i^*), \mathbf{x}_{i+j}^* - \mathbf{x}_i^* \rangle \right)$$
$$+ \frac{1}{i+j} \sum_{k=i+1}^{i+j} \left( f_k(\mathbf{x}_i^*, \mathbf{y}_{i+j}^*) - f_k(\mathbf{x}_{i+j}^*, \mathbf{y}_i^*) \right)$$
$$\leq \frac{i}{i+j} \left( -\frac{\mu}{2} \left\| \mathbf{y}_{i+j}^* - \mathbf{y}_i^* \right\|^2 - \frac{\mu}{2} \left\| \mathbf{x}_{i+j}^* - \mathbf{x}_i^* \right\|^2 \right) + \frac{1}{i+j} \sum_{k=i+1}^{i+j} \left( f_k(\mathbf{x}_i^*, \mathbf{y}_{i+j}^*) - f_k(\mathbf{x}_{i+j}^*, \mathbf{y}_i^*) \right).$$

The second inequality holds due to $\mu$-strong convexity in Assumption 3.7. The last inequality follows from the optimality of $\mathbf{y}_i^*$ and $\mathbf{x}_i^*$. Rearrange the terms, we have

$$\frac{2i+j}{i+j} \left( \frac{\mu}{2} \left\| \mathbf{x}_{i+j}^* - \mathbf{x}_i^* \right\|^2 + \frac{\mu}{2} \left\| \mathbf{y}_{i+j}^* - \mathbf{y}_i^* \right\|^2 \right)$$
$$\leq \frac{1}{i+j} \sum_{k=i+1}^{i+j} \left( f_k(\mathbf{x}_i^*, \mathbf{y}_{i+j}^*) - f_k(\mathbf{x}_i^*, \mathbf{y}_i^*) \right) + \frac{1}{i+j} \sum_{k=i+1}^{i+j} \left( f_k(\mathbf{x}_i^*, \mathbf{y}_i^*) - f_k(\mathbf{x}_{i+j}^*, \mathbf{y}_i^*) \right)$$
$$\leq \frac{jG \left( \left\| \mathbf{x}_{i+j}^* - \mathbf{x}_i^* \right\| + \left\| \mathbf{y}_{i+j}^* - \mathbf{y}_i^* \right\| \right)}{i+j}.$$

The last inequality follows from $G$-Lipschitzness of the function $f_k$ in Assumption 3.7. Multiply both sides of the inequality by 2, we have

$$\frac{2i+j}{i+j}\left(\mu\left\|\mathbf{x}_{i+j}^* - \mathbf{x}_i^*\right\|^2 + \mu\left\|\mathbf{y}_{i+j}^* - \mathbf{y}_i^*\right\|^2\right)$$

$$\leq \frac{2jG(\left\|\mathbf{x}_{i+j}^* - \mathbf{x}_i^*\right\| + \left\|\mathbf{y}_{i+j}^* - \mathbf{y}_i^*\right\|)}{i+j}$$

$$= \frac{2jG\left(\sqrt{\mu}\left\|\mathbf{x}_{i+j}^* - \mathbf{x}_i^*\right\| + \sqrt{\mu}\left\|\mathbf{y}_{i+j}^* - \mathbf{y}_i^*\right\|\right)}{\sqrt{\mu}(i+j)}$$

$$\leq \frac{2\sqrt{2}jG\sqrt{\mu\left\|\mathbf{x}_{i+j}^* - \mathbf{x}_i^*\right\|^2 + \mu\left\|\mathbf{y}_{i+j}^* - \mathbf{y}_i^*\right\|^2}}{\sqrt{\mu}(i+j)}.$$

The last inequality follows from $(a+b)^2 \leq 2a^2 + 2b^2$ for any $a, b \geq 0$. Noting that $\mathbf{z}_{i+j}^* = (\mathbf{x}_{i+j}^*, \mathbf{y}_{i+j}^*)$ and $\mathbf{z}_i^* = (\mathbf{x}_i^*, \mathbf{y}_i^*)$, we have

$$\sqrt{\mu\left\|\mathbf{z}_{i+j}^* - \mathbf{z}_i^*\right\|^2} = \sqrt{\mu\left\|\mathbf{x}_{i+j}^* - \mathbf{x}_i^*\right\|^2 + \mu\left\|\mathbf{y}_{i+j}^* - \mathbf{y}_i^*\right\|^2} \leq \frac{2\sqrt{2}jG}{\sqrt{\mu}(2i+j)}.$$

$\square$

The above lemma implies the following result.

**Lemma B.2.** *Let $\hat{\mathbf{z}}_j \in \mathcal{X} \times \mathcal{Y}$ be the approximate solution at stage $j \in [n]$. Then for all $i \in \{j+1, \ldots, n\}$,*

$$\mu\left\|\hat{\mathbf{z}}_j - \mathbf{z}_i^*\right\|^2 \leq \frac{16(G(i-j))^2}{\mu(i+j)^2} + 2\mu\left\|\hat{\mathbf{z}}_j - \mathbf{z}_j^*\right\|^2.$$

*Proof.* Applying Young's Inequality, we obtain

$$\mu\left\|\hat{\mathbf{z}}_j - \mathbf{z}_i^*\right\|^2$$

$$\leq 2\mu\left(\left\|\mathbf{z}_j^* - \mathbf{z}_i^*\right\|^2 + \left\|\mathbf{z}_j^* - \hat{\mathbf{z}}_j\right\|^2\right)$$

$$\leq \frac{16(G(i-j))^2}{\mu(i+j)^2} + 2\mu\left\|\hat{\mathbf{z}}_j - \mathbf{z}_j^*\right\|^2.$$

The last inequality follows from Lemma B.1. $\square$

Before studying the stochastic gradient estimator of CSVRG, we first show some properties about the gradient estimator $\tilde{\nabla}$.

**Lemma B.3.** *Let $\mathrm{prev}$ be the stage index at which a full prefix-sum gradient is computed, then*

$$\tilde{\nabla}_{i-1} = \frac{1}{i-1}\sum_{j=1}^{i-1} h_j(\hat{\mathbf{z}}_{\mathrm{prev}}). \tag{9}$$

*Proof.* We will prove the lemma by induction. Note that after stage $i = 1$, we set $\tilde{\nabla}_1 = h_1(\hat{\mathbf{z}}_1)$.

**Induction Hypothesis** Let Eq. (9) holds for stage $i - 1$.

**Induction Step** We will prove Eq. (9) holds for stage $i$. We consider three cases as below.

**(a).** $i - \text{prev} \geq \alpha i$. In this case, we have

$$\tilde{\nabla}_{i-1} = \frac{1}{i-1} \sum_{j=1}^{i-1} h_j(\hat{\mathbf{z}}_{i-1}),$$

In addition, we set $\text{prev} = i - 1$ and $h_j$ is defined in definition 3.5, therefore Eq. (9) holds at the current iteration.

**(b).** $(i-1) - \text{prev} \geq \alpha(i-1)$. It means that prev was updated in the previous stage. In this case, we infer from Algorithm 1 such that

$$\tilde{\nabla}_{i-1} = \frac{1}{i-1} \sum_{j=1}^{i-1} h_j(\hat{\mathbf{z}}_{i-1}).$$

Since we set $\text{prev} = i - 1$ in the last stage, Eq. (9) holds.

**(c).** $i - \text{prev} < \alpha i$ and $(i-1) - \text{prev} < \alpha(i-1)$. From the inductive hypothesis, we have

$$\tilde{\nabla}_{i-2} = \frac{1}{i-2} \sum_{j=1}^{i-2} \nabla h_j(\hat{\mathbf{z}}_{\text{prev}}). \tag{10}$$

Consequently, we reach step 17 of Algorithm 1, and it follows that

$$\begin{aligned}
\tilde{\nabla}_{i-1} &= \left(1 - \frac{1}{i-1}\right) \tilde{\nabla}_{i-2} + \frac{1}{i-1} h_{i-1}(\hat{\mathbf{z}}_{\text{prev}}) \\
&= \frac{1}{i-1} \sum_{j=1}^{i-2} h_j(\hat{\mathbf{z}}_{\text{prev}}) + \frac{1}{i-1} h_{i-1}(\hat{\mathbf{z}}_{\text{prev}}) \\
&= \frac{1}{i-1} \sum_{j=1}^{i-1} h_j(\hat{\mathbf{z}}_{\text{prev}}).
\end{aligned}$$

$\square$

Now we can show the stochastic gradient estimator $\bar{\nabla}$ is an unbiased gradient estimator of the prefix-sum objective function and it has bounded variance.

## B.2 PROOF OF LEMMA 5.1

*Proof.* According to the definition of $\bar{\nabla}$, one has

$$\begin{aligned}
\mathbb{E}[\bar{\nabla}] &= \left(1 - \frac{1}{i}\right) \left( \mathbb{E}\left[h_{u_t}(\mathbf{z}) - h_{u_t}(\hat{\mathbf{z}}_{\text{prev}})\right] + \tilde{\nabla}_{i-1} \right) + \frac{1}{i} h_i(\mathbf{z}) \\
&= \frac{i-1}{i} \left( \frac{1}{i-1} \sum_{j=1}^{i-1} (h_j(\mathbf{z}) - h_j(\hat{\mathbf{z}}_{\text{prev}})) + \tilde{\nabla}_{i-1} \right) + \frac{1}{i} h_i(\mathbf{z}) \\
&= \frac{1}{i} \sum_{j=1}^{i-1} h_j(\mathbf{z}) + \frac{1}{i} h_i(\mathbf{z}) \\
&= \frac{1}{i} \sum_{j=1}^{i} h_j(\mathbf{z}),
\end{aligned}$$

where we use the result of Lemma B.3 for the third equality. $\square$

### B.3 PROOF OF LEMMA 5.2

*Proof.* By the definition of $\bar{\nabla}$, we have

$$\mathbb{E}\left[\left\|\bar{\nabla} - \frac{1}{i}\sum_{j=1}^{i} h_j(\mathbf{z})\right\|^2\right]$$

$$= \left(1 - \frac{1}{i}\right)^2 \mathbb{E}\left[\left\|h_{u_t}(\mathbf{z}) - h_{u_t}(\hat{\mathbf{z}}_{\text{prev}}) + \tilde{\nabla}_{i-1} - \frac{1}{i-1}\sum_{j=1}^{i-1} h_j(\mathbf{z})\right\|^2\right]$$

$$\leq \left(1 - \frac{1}{i}\right)^2 \mathbb{E}\left[\left\|(h_{u_t}(\mathbf{z}) - h_{u_t}(\hat{\mathbf{z}}_{\text{prev}}))\right\|^2\right]$$

$$\leq \left(1 - \frac{1}{i}\right)^2 L^2 \mathbb{E}\left[\left\|\mathbf{z} - \hat{\mathbf{z}}_{\text{prev}}\right\|^2\right]$$

$$\leq \left(1 - \frac{1}{i}\right)^2 2L^2 \mathbb{E}\left[\left\|\mathbf{z} - \mathbf{z}_i^*\right\|^2 + \left\|\mathbf{z}_i^* - \hat{\mathbf{z}}_{\text{prev}}\right\|^2\right].$$

The first inequality follows from $\mathbb{E}[\|\mathbf{x} - \mathbb{E}[\mathbf{x}]\|^2] \leq \mathbb{E}[\|\mathbf{x}\|^2]$ for any random variable $\mathbf{x}$. The second inequality is derived using Assumption 3.7. Lemma B.2 implies that

$$\mathbb{E}\left[\left\|\hat{\mathbf{z}}_{\text{prev}} - \mathbf{z}_i^*\right\|^2\right]$$

$$= \frac{1}{\mu}\mathbb{E}\left[\mu\left\|\hat{\mathbf{z}}_{\text{prev}} - \mathbf{z}_i^*\right\|^2\right]$$

$$\leq \frac{1}{\mu}\mathbb{E}\left[\frac{16(G(i - \text{prev}))^2}{\mu(i + \text{prev})^2} + 2\mu\left\|\hat{\mathbf{z}}_{\text{prev}} - \mathbf{z}_{\text{prev}}^*\right\|^2\right]$$

$$\leq \frac{1}{\mu}\mathbb{E}\left[\frac{16(G(i - \text{prev}))^2}{\mu i^2} + 2\mu\left\|\hat{\mathbf{z}}_{\text{prev}} - \mathbf{z}_{\text{prev}}^*\right\|^2\right]$$

$$\leq \frac{16G^2\alpha^2}{\mu^2} + \frac{2}{\mu}\mathbb{E}\left[\mu\left\|\hat{\mathbf{z}}_{\text{prev}} - \mathbf{z}_{\text{prev}}^*\right\|^2\right].$$

The second inequality is due to the condition for full prefix-sum gradient computation in Algorithm 1. Consequently, one has

$$\mathbb{E}\left[\left\|\bar{\nabla} - \frac{1}{i}\sum_{j=1}^{i} h_j(\mathbf{z})\right\|^2\right]$$

$$\leq 2L^2\mathbb{E}\left[\left\|\mathbf{z} - \mathbf{z}_i^*\right\|^2\right] + \frac{32G^2L^2\alpha^2}{\mu^2} + \frac{4L^2}{\mu}\mathbb{E}\left[\mu\left\|\hat{\mathbf{z}}_{\text{prev}} - \mathbf{z}_{\text{prev}}^*\right\|^2\right].$$

$$\square$$

Now we can characterize the convergence rate of the CSVRG method at each stage with the following lemma.

**Lemma B.4.** *We suppose that $\mathbb{E}\left[\mu\left\|\hat{\mathbf{z}}_j - \mathbf{z}_j^*\right\|^2\right] \leq \epsilon$ holds for $j \in [i-1]$, running the CSVRG methods (Algorithm 1 with Option I) with $\gamma_t = \frac{2}{\mu(t+\beta)}$ where $\beta = \frac{6L^2}{\mu^2}$, then the output $\hat{\mathbf{z}}_i$ holds that*

$$\mathbb{E}\left[\mu\left\|\hat{\mathbf{z}}_i - \mathbf{z}_i^*\right\|^2\right] \leq \frac{(\beta-1)(\beta-2)}{(T_i+\beta-1)(T_i+\beta-2)}\mathbb{E}\left[\mu\left\|\mathbf{z}_i^0 - \mathbf{z}_i^*\right\|^2\right] + \frac{128G^2L^2\alpha^2}{\mu^3 T_i} + \frac{16L^2\epsilon}{\mu^2 T_i}.$$

*Proof.* Observe that $\mathbf{z}_i^* = \Pi_{\mathcal{X} \times \mathcal{Y}} \left( \mathbf{z}_i^* - \frac{\gamma_t}{i} \sum_{j=1}^i h_j(\mathbf{z}_i^*) \right)$, we have

$$
\left\| \mathbf{z}_i^{t+1} - \mathbf{z}_i^* \right\|^2
$$

$$
= \left\| \Pi_{\mathcal{X} \times \mathcal{Y}} (\mathbf{z}_i^t - \gamma_t \nabla_i^t) - \Pi_{\mathcal{X} \times \mathcal{Y}} \left( \mathbf{z}_i^* - \frac{\gamma_t}{i} \sum_{j=1}^i h_j(\mathbf{z}_i^*) \right) \right\|^2
$$

$$
\leq \left\| \mathbf{z}_i^t - \gamma_t \nabla_i^t - \mathbf{z}_i^* + \frac{\gamma_t}{i} \sum_{j=1}^i h_j(\mathbf{z}_i^*) \right\|^2
$$

$$
= \left\| \mathbf{z}_i^t - \mathbf{z}_i^* - \frac{\gamma_t}{i} \sum_{j=1}^i \left( h_j(\mathbf{z}_i^t) - h_j(\mathbf{z}_i^*) \right) - \gamma_t \left( \nabla_i^t - \frac{1}{i} \sum_{j=1}^i h_j(\mathbf{z}_i^t) \right) \right\|^2 .
$$

Taking expectations and multiplying $\mu$ on both sides of the inequality, we have

$$
\mathbb{E} \left[ \mu \left\| \mathbf{z}_i^{t+1} - \mathbf{z}_i^* \right\|^2 \right]
$$

$$
\leq \mathbb{E} \left[ \mu \left\| \mathbf{z}_i^t - \mathbf{z}_i^* \right\|^2 - \frac{2\mu\gamma_t}{i} \sum_{j=1}^i \langle h_j(\mathbf{z}_i^t) - h_j(\mathbf{z}_i^*), \mathbf{z}_i^t - \mathbf{z}_i^* \rangle + \frac{\mu\gamma_t^2}{i} \sum_{j=1}^i \left\| h_j(\mathbf{z}_i^t) - h_j(\mathbf{z}_i^*) \right\|^2 \right]
$$

$$
+ \mu\gamma_t^2 \mathbb{E} \left[ \left\| \nabla_i^t - \frac{1}{i} \sum_{j=1}^i h_j(\mathbf{z}_i^t) \right\|^2 \right] .
$$

Lemma A.4 implies that

$$
\mathbb{E} \left[ \mu \left\| \mathbf{z}_i^{t+1} - \mathbf{z}_i^* \right\|^2 \right]
$$

$$
\leq (1 - 2\mu\gamma_t) \mathbb{E} \left[ \mu \left\| \mathbf{z}_i^t - \mathbf{z}_i^* \right\|^2 \right] + \mu\gamma_t^2 L^2 \mathbb{E} \left[ \left\| \mathbf{z}_i^t - \mathbf{z}_i^* \right\|^2 \right] + \mu\gamma_t^2 \mathbb{E} \left[ \left\| \nabla_i^t - \frac{1}{i} \sum_{j=1}^i h_j(\mathbf{z}_i^t) \right\|^2 \right]
$$

$$
= (1 - 2\mu\gamma_t + \gamma_t^2 L^2) \mathbb{E} \left[ \mu \left\| \mathbf{z}_i^t - \mathbf{z}_i^* \right\|^2 \right] + \mu\gamma_t^2 \mathbb{E} \left[ \left\| \nabla_i^t - \frac{1}{i} \sum_{j=1}^i h_j(\mathbf{z}_i^t) \right\|^2 \right] .
$$

By applying Lemma 5.2, we obtain

$$
\mathbb{E} \left[ \mu \left\| \mathbf{z}_i^{t+1} - \mathbf{z}_i^* \right\|^2 \right]
$$

$$
\leq (1 - 2\mu\gamma_t + 3\gamma_t^2 L^2) \mathbb{E} \left[ \mu \left\| \mathbf{z}_i^t - \mathbf{z}_i^* \right\|^2 \right] + \frac{32 G^2 L^2 \alpha^2 \gamma_t^2}{\mu} + 4 L^2 \gamma_t^2 \mathbb{E} \left[ \mu \left\| \hat{\mathbf{z}}_{\text{prev}} - \mathbf{z}_{\text{prev}}^* \right\|^2 \right] .
$$

If we choose $\gamma_t = \frac{2}{\mu(t+\beta)}$, where $\beta = \frac{6L^2}{\mu^2}$, then

$$
3\gamma_t L^2 = \frac{6L^2}{\mu(t+\beta)} \leq \frac{6L^2}{\mu\beta} = \mu.
$$

Consequently, it follows that

$$
\mathbb{E} \left[ \mu \left\| \mathbf{z}_i^{t+1} - \mathbf{z}_i^* \right\|^2 \right]
$$

$$
\leq \frac{t+\beta-2}{t+\beta} \mathbb{E} \left[ \mu \left\| \mathbf{z}_i^t - \mathbf{z}_i^* \right\|^2 \right] + \frac{128 G^2 L^2 \alpha^2}{\mu^3 (t+\beta)^2} + \frac{16 L^2}{\mu^2 (t+\beta)^2} \mathbb{E} \left[ \mu \left\| \hat{\mathbf{z}}_{\text{prev}} - \mathbf{z}_{\text{prev}}^* \right\|^2 \right] .
$$

Multiply both sides by $(t + \beta)(t + \beta - 1)$, we obtain

$$(t + \beta)(t + \beta - 1)\mathbb{E}\left[\mu \left\|\mathbf{z}_i^{t+1} - \mathbf{z}_i^*\right\|^2\right]$$

$$\leq (t + \beta - 1)(t + \beta - 2)\mathbb{E}\left[\mu \left\|\mathbf{z}_i^t - \mathbf{z}_i^*\right\|^2\right] + \frac{128G^2L^2\alpha^2}{\mu^3} + \frac{16L^2}{\mu^2}\mathbb{E}\left[\mu \left\|\hat{\mathbf{z}}_{\text{prev}} - \mathbf{z}_{\text{prev}}^*\right\|^2\right]$$

$$\leq (t + \beta - 1)(t + \beta - 2)\mathbb{E}\left[\mu \left\|\mathbf{z}_i^t - \mathbf{z}_i^*\right\|^2\right] + \frac{128G^2L^2\alpha^2}{\mu^3} + \frac{16L^2\epsilon}{\mu^2}.$$

The last inequality is due to the induction hypothesis that $\mathbb{E}\left[\mu \left\|\hat{\mathbf{z}}_{\text{prev}} - \mathbf{z}_{\text{prev}}^*\right\|^2\right] \leq \epsilon$. By summing up the above equality for $t$ from 0 to $T_i - 1$, we have

$$(T_i + \beta - 1)(T_i + \beta - 2)\mathbb{E}\left[\mu \left\|\mathbf{z}_i^{T_i} - \mathbf{z}_i^*\right\|^2\right]$$

$$\leq (\beta - 1)(\beta - 2)\mathbb{E}\left[\mu \left\|\mathbf{z}_i^0 - \mathbf{z}_i^*\right\|^2\right] + \frac{128G^2L^2\alpha^2 T_i}{\mu^3} + \frac{16L^2\epsilon T_i}{\mu^2}.$$

Divide both sides of the above inequality by $(T_i + \beta - 1)(T_i + \beta - 2)$, one has

$$\mathbb{E}\left[\mu \left\|\mathbf{z}_i^{T_i} - \mathbf{z}_i^*\right\|^2\right]$$

$$\leq \frac{(\beta - 1)(\beta - 2)}{(T_i + \beta - 1)(T_i + \beta - 2)}\mathbb{E}\left[\mu \left\|\mathbf{z}_i^0 - \mathbf{z}_i^*\right\|^2\right] + \frac{128G^2L^2\alpha^2}{\mu^3 T_i} + \frac{16L^2\epsilon}{\mu^2 T_i}.$$

The inequality holds due to the observation $\beta \geq 1$. Noting that $\hat{\mathbf{z}}_i = \mathbf{z}_i^{T_i}$, we get the desired bound. $\qquad\square$

After characterizing the convergence rate of the CSVRG of a single stage, we wish to compute the total time complexity of the CSVRG method. The following two lemmas are instrumental in establishing the total complexity of the CSVRG method.

**Lemma B.5** (Mavrothalassitis et al. (2024)). *Over a sequence of $n$ stages in Algorithm 1, the condition $i - \text{prev} \geq \alpha i$ is satisfied for $\lceil \log n/\alpha \rceil$ times.*

**Corollary B.6** (Mavrothalassitis et al. (2024)). *Over a sequence of $n$ stages in Algorithm 1, it requires $\mathcal{O}(\sum_{i=1}^n T_i + n\lceil \log n/\alpha \rceil)$ FOs.*

We can show that under appropriate choices of hyperparameters, the CSVRG method obtain a sequence of $\epsilon$-saddle points for the continual minimax optimization problem.

### B.4 PROOF OF THEOREM 5.3

*Proof.* We prove the theorem by induction.
**Induction Hypothesis.** At epoch 1, Algorithm 1 performs extragradient on $f_1$ to produce $(\hat{\mathbf{x}}_1, \hat{\mathbf{y}}_1)$, and we obtain

$$\mathbb{E}[\mu \left\|\hat{\mathbf{z}}_1 - \mathbf{z}_1^*\right\|^2] \leq \epsilon.$$

with $\mathcal{O}(L/\mu \log(1/\epsilon))$ FOs. Now we assume

$$\mathbb{E}\left[\mu \left\|\hat{\mathbf{z}}_j - \mathbf{z}_j^*\right\|^2\right] \leq \epsilon$$

holds for epochs $j \in [i - 1]$.

**Induction Step.** By Lemma B.2, we have

$$\mu \left\|\mathbf{z}_i^0 - \mathbf{z}_i^*\right\|^2$$

$$= \mu \left\|\hat{\mathbf{z}}_{i-1} - \mathbf{z}_i^*\right\|^2$$

$$\leq \frac{16G^2}{\mu(2i-1)^2} + 2\mu\mathbb{E}\left[\left\|\hat{\mathbf{z}}_{i-1} - \mathbf{z}_{i-1}^*\right\|^2\right]$$

$$\leq \frac{16G^2}{\mu(2i-1)^2} + 2\epsilon.$$

The last inequality is due to the induction hypothesis. By Lemma B.4, we have

$$\mathbb{E}\left[\mu\left\|\hat{\mathbf{z}}_i - \mathbf{z}_i^*\right\|^2\right]$$

$$\leq \frac{\beta^2}{T_i^2}\mathbb{E}\left[\mu\left\|\mathbf{z}_i^0 - \mathbf{z}_i^*\right\|^2\right] + \frac{128G^2L^2\alpha^2}{\mu^3T_i} + \frac{16L^2\epsilon}{\mu^2T_i}$$

$$= \frac{36L^4}{\mu^4T_i^2}\mathbb{E}\left[\mu\left\|\mathbf{z}_i^0 - \mathbf{z}_i^*\right\|^2\right] + \frac{128G^2L^2\alpha^2}{\mu^3T_i} + \frac{16L^2\epsilon}{\mu^2T_i}$$

$$\leq \frac{144L^4G^2}{\mu^5T_i^2i^2} + \frac{72L^4\epsilon}{\mu^4T_i^2} + \frac{128G^2L^2\alpha^2}{\mu^3T_i} + \frac{16L^2\epsilon}{\mu^2T_i}.$$

By taking $\alpha = \frac{\mu\epsilon^{\frac{1}{3}}}{G^{\frac{2}{3}}L^{\frac{2}{3}}}$, and we choose

$$T_i = \mathcal{O}\left(\frac{L^2G}{\mu^{\frac{5}{2}}i\epsilon^{\frac{1}{2}}} + \frac{L^2}{\mu^2} + \frac{G^{\frac{2}{3}}L^{\frac{2}{3}}}{\mu\epsilon^{\frac{1}{3}}}\right),$$

then it holds that

$$\mathbb{E}\left[\mu\left\|\hat{\mathbf{z}}_i - \mathbf{z}_i^*\right\|^2\right] \leq \epsilon.$$

$\square$

Accordingly, the total time complexity of the CSVRG method is presented in the following corollary.

### B.5 PROOF OF COROLLARY 5.4

*Proof.* By adding up $T_i$ from $i = 1$ to $n$, we have

$$\sum_{i=1}^n T_i = \mathcal{O}\left(\frac{L^2G\log n}{\mu^{\frac{5}{2}}\epsilon^{\frac{1}{2}}} + \frac{nL^2}{\mu^2} + \frac{nG^{\frac{2}{3}}L^{\frac{2}{3}}}{\mu\epsilon^{\frac{1}{3}}}\right).$$

In addition, recall that we choose $\alpha = \frac{\mu\epsilon^{\frac{1}{3}}}{G^{\frac{2}{3}}L^{\frac{2}{3}}}$, then

$$n\log n/\alpha = \mathcal{O}\left(\frac{nG^{\frac{2}{3}}L^{\frac{2}{3}}\log n}{\mu\epsilon^{\frac{1}{3}}}\right).$$

Corollary B.6 implies the total IFO calls. $\square$

## C CONVERGENCE ANALYSIS OF THE CSVRE METHOD

In this section, we present the convergence analysis of the CSVRE method for solving the continual minimax optimization problem in the strongly-convex-strongly-concave case. We first show the convergence of the CSVRE method at each stage as follows.

**Lemma C.1.** *We suppose that $\mathbb{E}[\mu\left\|\hat{\mathbf{z}}_j - \mathbf{z}_j^*\right\|^2] \leq \epsilon$ holds for $j \in [i-1]$, running the CSVRE methods (Algorithm 1 with Option II) with $\gamma_t = \frac{2}{\mu(t+\beta)}$ where $\beta = 8L/\mu$, and $\eta_t = 1 - 2\gamma_t^2L^2$, then the output $\hat{\mathbf{z}}_i$ holds that*

$$\mu\left\|\hat{\mathbf{z}}_i - \mathbf{z}_i^*\right\|^2 \leq \frac{72L^2\left\|\mathbf{z}_i^0 - \mathbf{z}_i^*\right\|^2}{\mu T_i^2} + \frac{144L^2G^2\alpha^2}{\mu^3T_i} + \frac{18L^2\epsilon}{\mu^2T_i}.$$

*Proof.* Let $\tilde{h}_i(\mathbf{z}) = \frac{1}{i}\sum_{j=1}^i h_j(\mathbf{z})$, the proximal update of $\mathbf{z}_i^{t+1/2}$ and $\mathbf{z}_i^{t+1}$ and Lemma A.2 imply that

$$\left\langle \mathbf{z}_i^{t+1/2} - \bar{\mathbf{z}}_i^t + \gamma_t\left(\left(1 - \frac{1}{i}\right)\tilde{h}_{i-1}(\hat{\mathbf{z}}_{\text{prev}}) + \frac{1}{i}h_i(\hat{\mathbf{z}}_{\text{prev}})\right), \mathbf{z}_i^{t+1} - \mathbf{z}_i^{t+1/2}\right\rangle \geq 0,$$

$$\left\langle \mathbf{z}_i^{t+1} - \bar{\mathbf{z}}_i^t + \gamma_t\left(\left(1 - \frac{1}{i}\right)\left(h_{u_t}(\mathbf{z}_i^{t+1/2}) - h_{u_t}(\hat{\mathbf{z}}_{\text{prev}}) + \tilde{h}_{i-1}(\hat{\mathbf{z}}_{\text{prev}})\right) + \frac{1}{i}h_i(\mathbf{z}_i^{t+1/2})\right), \mathbf{z}_i^* - \mathbf{z}_i^{t+1}\right\rangle \geq 0.$$

Summing up these two inequalities, we have

$$
\begin{aligned}
0 \leq &\langle \mathbf{z}_i^{t+1/2} - \bar{\mathbf{z}}_i^t, \mathbf{z}_i^{t+1} - \mathbf{z}_i^{t+1/2}\rangle + \langle \mathbf{z}_i^{t+1} - \bar{\mathbf{z}}_i^t, \mathbf{z}_i^* - \mathbf{z}_i^{t+1}\rangle + \\
&\gamma_t \left\langle \left(1 - \frac{1}{i}\right)\left(h_{u_t}(\mathbf{z}_i^{t+1/2}) - h_{u_t}(\hat{\mathbf{z}}_{\text{prev}}) + \tilde{h}_{i-1}(\hat{\mathbf{z}}_{\text{prev}})\right) + \frac{1}{i} h_i(\mathbf{z}_i^{t+1/2}), \mathbf{z}_i^* - \mathbf{z}_i^{t+1/2}\right\rangle + \\
&\gamma_t \left\langle \left(1 - \frac{1}{i}\right)\left(h_{u_t}(\mathbf{z}_i^{t+1/2}) - h_{u_t}(\hat{\mathbf{z}}_{\text{prev}})\right) + \frac{1}{i}\left(h_i(\mathbf{z}_i^{t+1/2}) - h_i(\hat{\mathbf{z}}_{\text{prev}})\right), \mathbf{z}_i^{t+1/2} - \mathbf{z}_i^{t+1}\right\rangle.
\end{aligned}
\tag{11}
$$

Using $2\langle \mathbf{a}, \mathbf{b}\rangle = \|\mathbf{a} + \mathbf{b}\|^2 - \|\mathbf{a}\|^2 - \|\mathbf{b}\|^2$ and recall that $\bar{\mathbf{z}}_i^t = \eta_t \mathbf{z}_i^t + (1 - \eta_t)\hat{\mathbf{z}}_{\text{prev}}$, the first term on the r.h.s. of (11) can be written as

$$
\begin{aligned}
&2\left\langle \mathbf{z}_i^{t+1/2} - \bar{\mathbf{z}}_i^t, \mathbf{z}_i^{t+1} - \mathbf{z}_i^{t+1/2}\right\rangle \\
=&2\eta_t \left\langle \mathbf{z}_i^{t+1/2} - \mathbf{z}_i^t, \mathbf{z}_i^{t+1} - \mathbf{z}_i^{t+1/2}\right\rangle + 2(1 - \eta_t)\left\langle \mathbf{z}_i^{t+1/2} - \hat{\mathbf{z}}_{\text{prev}}, \mathbf{z}_i^{t+1} - \mathbf{z}_i^{t+1/2}\right\rangle \\
=&\eta_t \left\|\mathbf{z}_i^t - \mathbf{z}_i^{t+1}\right\|^2 - \eta_t \left\|\mathbf{z}_i^{t+1/2} - \mathbf{z}_i^t\right\|^2 + (1 - \eta_t)\left\|\hat{\mathbf{z}}_{\text{prev}} - \mathbf{z}_i^{t+1}\right\|^2 \\
&- (1 - \eta_t)\left\|\mathbf{z}_i^{t+1/2} - \hat{\mathbf{z}}_{\text{prev}}\right\|^2 - \left\|\mathbf{z}_i^{t+1} - \mathbf{z}_i^{t+1/2}\right\|^2.
\end{aligned}
\tag{12}
$$

Similarly, the second term of (11) can be written as

$$
\begin{aligned}
&2\left\langle \mathbf{z}_i^{t+1} - \bar{\mathbf{z}}_i^t, \mathbf{z}_i^* - \mathbf{z}_i^{t+1}\right\rangle \\
=&2\left\langle \mathbf{z}_i^{t+1} - \eta_t \mathbf{z}_i^t - (1 - \eta_t)\hat{\mathbf{z}}_{\text{prev}}, \mathbf{z}_i^* - \mathbf{z}_i^{t+1}\right\rangle \\
=&2\eta_t \left\langle \mathbf{z}_i^{t+1} - \mathbf{z}_i^t, \mathbf{z}_i^* - \mathbf{z}_i^{t+1}\right\rangle + 2(1 - \eta_t)\left\langle \mathbf{z}_i^{t+1} - \hat{\mathbf{z}}_{\text{prev}}, \mathbf{z}_i^* - \mathbf{z}_i^{t+1}\right\rangle \\
=&\eta_t \left(\left\|\mathbf{z}_i^t - \mathbf{z}_i^*\right\|^2 - \left\|\mathbf{z}_i^{t+1} - \mathbf{z}_i^t\right\|^2 - \left\|\mathbf{z}_i^* - \mathbf{z}_i^{t+1}\right\|^2\right) \\
&+ (1 - \eta_t)\left(\left\|\hat{\mathbf{z}}_{\text{prev}} - \mathbf{z}_i^*\right\|^2 - \left\|\mathbf{z}_i^{t+1} - \hat{\mathbf{z}}_{\text{prev}}\right\|^2 - \left\|\mathbf{z}_i^* - \mathbf{z}_i^{t+1}\right\|^2\right) \\
=&\eta_t \left\|\mathbf{z}_i^t - \mathbf{z}_i^*\right\|^2 - \eta_t \left\|\mathbf{z}_i^{t+1} - \mathbf{z}_i^t\right\|^2 + (1 - \eta_t)\left\|\hat{\mathbf{z}}_{\text{prev}} - \mathbf{z}_i^*\right\|^2 \\
&- (1 - \eta_t)\left\|\mathbf{z}_i^{t+1} - \hat{\mathbf{z}}_{\text{prev}}\right\|^2 - \left\|\mathbf{z}_i^* - \mathbf{z}_i^{t+1}\right\|^2.
\end{aligned}
\tag{13}
$$

Using the fact $\mathbb{E}\left[h_{u_t}(\mathbf{z})\right] = \tilde{h}_{i-1}(\mathbf{z})$ for $\forall \mathbf{z} \in \mathcal{X} \times \mathcal{Y}$, we can bound the third term of (11) as follows

$$
\begin{aligned}
&2\mathbb{E}\left[\left\langle \left(1 - \frac{1}{i}\right)\left(h_{u_t}(\mathbf{z}_i^{t+1/2}) - h_{u_t}(\hat{\mathbf{z}}_{\text{prev}}) + \tilde{h}_{i-1}(\hat{\mathbf{z}}_{\text{prev}})\right) + \frac{1}{i} h_i(\mathbf{z}_i^{t+1/2}), \mathbf{z}_i^* - \mathbf{z}_i^{t+1/2}\right\rangle\right] \\
=&2\left\langle \left(1 - \frac{1}{i}\right)\tilde{h}_{i-1}\left(\mathbf{z}_i^{t+1/2}\right) + \frac{1}{i} h_i\left(\mathbf{z}_i^{t+1/2}\right), \mathbf{z}_i^* - \mathbf{z}_i^{t+1/2}\right\rangle \\
=&2\left\langle \tilde{h}_i\left(\mathbf{z}_i^{t+1/2}\right), \mathbf{z}_i^* - \mathbf{z}_i^{t+1/2}\right\rangle \\
\leq&2\left\langle \tilde{h}_i\left(\mathbf{z}_i^*\right), \mathbf{z}_i^* - \mathbf{z}_i^{t+1/2}\right\rangle - 2\mu \left\|\mathbf{z}_i^* - \mathbf{z}_i^{t+1/2}\right\|^2 \\
\leq&-\mu \left\|\mathbf{z}_i^* - \mathbf{z}_i^{t+1}\right\|^2 + 2\mu \left\|\mathbf{z}_i^{t+1} - \mathbf{z}_i^{t+1/2}\right\|^2.
\end{aligned}
\tag{14}
$$

The last inequality is due to $\|\mathbf{a} + \mathbf{b}\|^2 \leq 2\|\mathbf{a}\|^2 + 2\|\mathbf{b}\|^2$ and the optimality of $\mathbf{z}_i^*$. Furthermore, using Young's inequality, we obtain

$$
\mathbb{E}\left[2\gamma_t \left\langle \left(1 - \frac{1}{i}\right)\left(h_{u_t}(\mathbf{z}_i^{t+1/2}) - h_{u_t}(\hat{\mathbf{z}}_{\mathrm{prev}})\right) + \frac{1}{i}\left(h_i(\mathbf{z}_i^{t+1/2}) - h_i(\hat{\mathbf{z}}_{\mathrm{prev}})\right), \mathbf{z}_i^{t+1/2} - \mathbf{z}_i^{t+1}\right\rangle\right]
$$

$$
=\mathbb{E}\left[2\gamma_t \left\langle \left(1 - \frac{1}{i}\right)\left(\tilde{h}_{i-1}(\mathbf{z}_i^{t+1/2}) - \tilde{h}_{i-1}(\hat{\mathbf{z}}_{\mathrm{prev}})\right) + \frac{1}{i}\left(h_i(\mathbf{z}_i^{t+1/2}) - h_i(\hat{\mathbf{z}}_{\mathrm{prev}})\right), \mathbf{z}_i^{t+1/2} - \mathbf{z}_i^{t+1}\right\rangle\right]
$$

$$
=\mathbb{E}\left[2\gamma_t \left\langle \tilde{h}_i(\mathbf{z}_i^{t+1/2}) - \tilde{h}_i(\hat{\mathbf{z}}_{\mathrm{prev}}), \mathbf{z}_i^{t+1/2} - \mathbf{z}_i^{t+1}\right\rangle\right]
$$

$$
\leq 2\gamma_t^2 \mathbb{E}\left[\left\|\tilde{h}_i(\mathbf{z}_i^{t+1/2}) - \tilde{h}_i(\hat{\mathbf{z}}_{\mathrm{prev}})\right\|^2\right] + \frac{1}{2}\mathbb{E}\left[\left\|\mathbf{z}_i^{t+1/2} - \mathbf{z}_i^{t+1}\right\|^2\right]
$$

$$
\leq 2\gamma_t^2 L^2 \mathbb{E}\left[\left\|\mathbf{z}_i^{t+1/2} - \hat{\mathbf{z}}_{\mathrm{prev}}\right\|^2\right] + \frac{1}{2}\mathbb{E}\left[\left\|\mathbf{z}_i^{t+1/2} - \mathbf{z}_i^{t+1}\right\|^2\right],
$$

$$
\tag{15}
$$

where the first inequality is due to Young's inequality and the last inequality follows from Assumption 3.7. Substituting (12)-(15) into eq. (11), we can show that

$$
\eta_t \left\|\mathbf{z}_i^t - \mathbf{z}_i^*\right\|^2 + (1 - \eta_t)\left\|\hat{\mathbf{z}}_{\mathrm{prev}} - \mathbf{z}_i^*\right\|^2 - \left\|\mathbf{z}_i^* - \mathbf{z}_i^{t+1}\right\|^2 - \eta_t \left\|\mathbf{z}_i^{t+1/2} - \mathbf{z}_i^t\right\|^2
$$

$$
- (1 - \eta_t)\left\|\mathbf{z}_i^{t+1/2} - \hat{\mathbf{z}}_{\mathrm{prev}}\right\|^2 - \left\|\mathbf{z}_i^{t+1} - \mathbf{z}_i^{t+1/2}\right\|^2
$$

$$
- \gamma_t \mu \left\|\mathbf{z}_i^* - \mathbf{z}_i^{t+1}\right\|^2 + 2\gamma_t \mu \left\|\mathbf{z}_i^{t+1} - \mathbf{z}_i^{t+1/2}\right\|^2
$$

$$
+ 2\gamma_t^2 L^2 \mathbb{E}\left[\left\|\mathbf{z}_i^{t+1/2} - \hat{\mathbf{z}}_{\mathrm{prev}}\right\|^2\right] + \frac{1}{2}\mathbb{E}\left[\left\|\mathbf{z}_i^{t+1/2} - \mathbf{z}_i^{t+1}\right\|^2\right] \geq 0.
$$

Rearrange the terms, we have

$$
(1 + \gamma_t \mu)\left\|\mathbf{z}_i^* - \mathbf{z}_i^{t+1}\right\|^2
$$

$$
\leq \eta_t \left\|\mathbf{z}_i^t - \mathbf{z}_i^*\right\|^2 + (1 - \eta_t)\left\|\hat{\mathbf{z}}_{\mathrm{prev}} - \mathbf{z}_i^*\right\|^2 - \left(\frac{1}{2} - 2\gamma_t \mu\right)\left\|\mathbf{z}_i^{t+1/2} - \mathbf{z}_i^{t+1}\right\|^2
$$

$$
- \left(1 - \eta_t - 2\gamma_t^2 L^2\right)\left\|\mathbf{z}_i^{t+1/2} - \hat{\mathbf{z}}_{\mathrm{prev}}\right\|^2.
$$

If we choose $\gamma_t = \frac{2}{\mu(t+\beta)}$ and $\eta_t = 1 - 2\gamma_t^2 L^2$ where $\beta = 8L/\mu$, such that

$$
\frac{1}{2} - 2\gamma_t \mu \geq 0, \quad 1 - 2\gamma_t^2 L^2 \geq 0.
$$

We can show that

$$
(1 + \gamma_t \mu)\left\|\mathbf{z}_i^* - \mathbf{z}_i^{t+1}\right\|^2
$$

$$
\leq \eta_t \left\|\mathbf{z}_i^t - \mathbf{z}_i^*\right\|^2 + (1 - \eta_t)\left\|\hat{\mathbf{z}}_{\mathrm{prev}} - \mathbf{z}_i^*\right\|^2
$$

$$
\leq \eta_t \left\|\mathbf{z}_i^t - \mathbf{z}_i^*\right\|^2 + (1 - \eta_t)\left[\frac{16(G(i - \mathrm{prev}))^2}{\mu^2(i + \mathrm{prev})^2} + 2\left\|\hat{\mathbf{z}}_{\mathrm{prev}} - \mathbf{z}_{\mathrm{prev}}^*\right\|^2\right]
$$

$$
\leq \eta_t \left\|\mathbf{z}_i^t - \mathbf{z}_i^*\right\|^2 + (1 - \eta_t)\left[\frac{16G^2 i^2 \alpha^2}{\mu^2 i^2} + 2\left\|\hat{\mathbf{z}}_{\mathrm{prev}} - \mathbf{z}_{\mathrm{prev}}^*\right\|^2\right]
$$

$$
\leq \eta_t \left\|\mathbf{z}_i^t - \mathbf{z}_i^*\right\|^2 + \frac{(1 - \eta_t)16G^2 \alpha^2}{\mu^2} + \frac{2(1 - \eta_t)\epsilon}{\mu}.
$$

The second inequality follows from Lemma B.2. The last inequality is due to the induction hypothesis. Since we choose $\gamma_t = \frac{2}{\mu(t+\beta)}$ and $\eta_t = 1 - 2\gamma_t^2 L^2$ where $\beta = 8L/\mu$, then

$$
\frac{t + \beta + 2}{t + \beta}\left\|\mathbf{z}_i^* - \mathbf{z}_i^{t+1}\right\|^2 \leq \left\|\mathbf{z}_i^t - \mathbf{z}_i^*\right\|^2 + \frac{128L^2 G^2 \alpha^2}{\mu^4(t+\beta)^2} + \frac{16L^2 \epsilon}{\mu^3(t+\beta)^2}.
$$

Multiply both sides by $(t + \beta)(t + \beta + 1)$, we have

$$(t + \beta + 2)(t + \beta + 1) \left\| \mathbf{z}_i^* - \mathbf{z}_i^{t+1} \right\|^2$$

$$\leq (t + \beta)(t + \beta + 1) \left\| \mathbf{z}_i^t - \mathbf{z}_i^* \right\|^2 + \frac{128 L^2 G^2 \alpha^2 (t + \beta + 1)}{\mu^4 (t + \beta)} + \frac{16 L^2 \epsilon (t + \beta + 1)}{\mu^3 (t + \beta)}.$$

Observe that

$$\frac{t + \beta + 1}{t + \beta} \leq \frac{\beta + 1}{\beta} \leq \frac{9}{8},$$

we can deduce that

$$(t + \beta + 2)(t + \beta + 1)\mu \left\| \mathbf{z}_i^* - \mathbf{z}_i^{t+1} \right\|^2$$

$$\leq (t + \beta)(t + \beta + 1)\mu \left\| \mathbf{z}_i^t - \mathbf{z}_i^* \right\|^2 + \frac{144 L^2 G^2 \alpha^2}{\mu^3} + \frac{18 L^2 \epsilon}{\mu^2}.$$

Sum up the above inequalities from $t = 0$ to $T_i - 1$, we can obtain that

$$(T_i + \beta + 1)(T_i + \beta)\mu \left\| \mathbf{z}_i^* - \mathbf{z}_i^{T_i} \right\|^2 \leq \beta(\beta + 1)\mu \left\| \mathbf{z}_i^0 - \mathbf{z}_i^* \right\|^2 + \frac{144 L^2 G^2 \alpha^2 T_i}{\mu^3} + \frac{18 L^2 \epsilon T_i}{\mu^2}.$$

Dividing both sides by $(T_i + \beta + 1)(T_i + \beta)$ and substituting $\beta = 8L/\mu$, one has

$$\mu \left\| \mathbf{z}_i^{T_i} - \mathbf{z}_i^* \right\|^2 \leq \frac{\beta(\beta + 1)\mu \left\| \mathbf{z}_i^0 - \mathbf{z}_i^* \right\|^2}{(T_i + \beta + 1)(T_i + \beta)} + \frac{144 L^2 G^2 \alpha^2 T_i}{\mu^3 (T_i + \beta + 1)(T_i + \beta)} + \frac{18 L^2 \epsilon T_i}{\mu^2 (T_i + \beta + 1)(T_i + \beta)}$$

$$\leq \frac{72 L^2 \left\| \mathbf{z}_i^0 - \mathbf{z}_i^* \right\|^2}{\mu T_i^2} + \frac{144 L^2 G^2 \alpha^2}{\mu^3 T_i} + \frac{18 L^2 \epsilon}{\mu^2 T_i}.$$

Noting $\hat{\mathbf{z}}_i = \mathbf{z}_i^{T_i}$, it completes the proof. $\qquad\square$

We can show that under appropriate choices of hyperparameters, the CSVRE method obtains a sequence of $\epsilon$-saddle points for the continual minimax optimization problem.

## C.1  PROOF OF THEOREM 5.5

*Proof.* We prove the theorem by induction.

**Induction Hypothesis.** At epoch 1, Algorithm 1 performs the extragradient method on $f_1$ to produce $(\hat{\mathbf{x}}_1, \hat{\mathbf{y}}_1)$, and we obtain

$$\mathbb{E}[\mu \left\| \hat{\mathbf{z}}_1 - \mathbf{z}_1^* \right\|^2] = \mathbb{E}[\mu \left\| \hat{\mathbf{x}}_1 - \mathbf{x}_1^* \right\|^2 + \mu \left\| \hat{\mathbf{y}}_1 - \mathbf{y}_1^* \right\|^2] \leq \epsilon.$$

with $\mathcal{O}(L/\mu \log(1/\epsilon))$ FOs. Now we assume $\mathbb{E}\left[ \mu \left\| \hat{\mathbf{z}}_j - \mathbf{z}_j^* \right\|^2 \right] \leq \epsilon$ holds for epochs $j \in [i - 1]$.

**Induction Step.** By Lemma B.2, we have

$$\mu \left\| \mathbf{z}_i^0 - \mathbf{z}_i^* \right\|^2$$

$$= \mu \left\| \hat{\mathbf{z}}_{i-1} - \mathbf{z}_i^* \right\|^2$$

$$\leq \frac{16 G^2}{\mu(2i - 1)^2} + 2\mu \left\| \hat{\mathbf{z}}_{i-1} - \mathbf{z}_{i-1}^* \right\|^2$$

$$\leq \frac{16 G^2}{\mu(2i - 1)^2} + 2\epsilon.$$

The last inequality is due to the induction hypothesis. By Lemma C.1, we have

$$\mu \left\| \hat{\mathbf{z}}_i - \mathbf{z}_i^* \right\|^2$$

$$\leq \frac{72 L^2}{\mu^2 T_i^2} \left( \frac{16 G^2}{\mu(2i - 1)^2} + 2\epsilon \right) + \frac{144 L^2 G^2 \alpha^2}{\mu^3 T_i} + \frac{18 L^2 \epsilon}{\mu^2 T_i}$$

$$= \frac{1152 L^2 G^2}{\mu^3 (2i - 1)^2 T_i^2} + \frac{144 L^2 \epsilon}{\mu^2 T_i^2} + \frac{144 L^2 G^2 \alpha^2}{\mu^3 T_i} + \frac{18 L^2 \epsilon}{\mu^2 T_i}.$$

By taking $\alpha = \frac{\mu\epsilon^{\frac{1}{3}}}{G^{\frac{2}{3}}L^{\frac{2}{3}}}$ and choose

$$T_i = \mathcal{O}\left(\frac{LG}{\mu^{\frac{3}{2}}i\epsilon^{\frac{1}{2}}} + \frac{L^2}{\mu^2} + \frac{G^{\frac{2}{3}}L^{\frac{2}{3}}}{\mu\epsilon^{\frac{1}{3}}}\right),$$

we obtain

$$\mu\left\|\hat{\mathbf{z}}_i - \mathbf{z}_i^*\right\|^2 \le \epsilon.$$

$\square$

### C.2 Proof of Corollary 5.6

*Proof.* By summing up $T_i$ from $i = 1$ to $n$, we have

$$\sum_{i=1}^{n} T_i = \mathcal{O}\left(\frac{LG\log n}{\mu^{\frac{3}{2}}\epsilon^{\frac{1}{2}}} + \frac{L^2 n}{\mu^2} + \frac{G^{\frac{2}{3}}L^{\frac{2}{3}}n}{\mu\epsilon^{\frac{1}{3}}}\right).$$

In addition, by choosing $\alpha = \frac{\mu\epsilon^{\frac{1}{3}}}{G^{\frac{2}{3}}L^{\frac{2}{3}}}$, one has

$$n\log n/\alpha = \mathcal{O}\left(\frac{nG^{\frac{2}{3}}L^{\frac{2}{3}}\log n}{\mu\epsilon^{\frac{1}{3}}}\right).$$

Applying Corollary B.6 implies the total IFO calls. $\square$

## D  Convergence Analysis of the General Convex-Concave Setting

This section extends the proposed stochastic algorithms in Section 4 to solve the continual finite-sum minimax optimization problem where each component function $f_i$ is convex in $\mathbf{x}$ and concave in $\mathbf{y}$. For the general convex-concave setting, we are interested in finding an approximate suboptimal solution in terms of the duality gap, which is defined as follows.

**Definition D.1.** *For the convex-concave minimax optimization problem $\min_{\mathbf{x}\in\mathcal{X}}\max_{\mathbf{y}\in\mathcal{Y}} f(\mathbf{x}, \mathbf{y})$, the point $(\hat{\mathbf{x}}, \hat{\mathbf{y}}) \in \mathcal{X} \times \mathcal{Y}$ is said to be an $\epsilon$-suboptimal solution w.r.t. the duality gap if*

$$\max_{\mathbf{y}\in\mathcal{Y}} f(\hat{\mathbf{x}}, \mathbf{y}) - \min_{\mathbf{x}\in\mathcal{X}} f(\mathbf{x}, \hat{\mathbf{y}}) \le \epsilon.$$

We introduce the following augmented function for each component function $f_i$,

$$f_{i,\epsilon}(\mathbf{x}, \mathbf{y}) := f_i(\mathbf{x}, \mathbf{y}) + \frac{\epsilon}{8D_{\mathcal{X}}^2}\left\|\mathbf{x} - \hat{\mathbf{x}}_0\right\|^2 - \frac{\epsilon}{8D_{\mathcal{Y}}^2}\left\|\mathbf{y} - \hat{\mathbf{y}}_0\right\|^2,$$

where $\hat{\mathbf{x}}_0 \in \mathcal{X}$ and $\hat{\mathbf{y}}_0 \in \mathcal{Y}$ are some initial points. We can infer that the function $f_{i,\epsilon}$ is $\Theta(\epsilon)$-strongly-convex-strongly-concave. We also define the corresponding prefix-sum objective function as $g_{i,\epsilon}(\mathbf{x}, \mathbf{y}) := \sum_{j=1}^{i} f_{j,\epsilon}(\mathbf{x}, \mathbf{y})/i$, and the gradient

$$h_{i,\epsilon}(\mathbf{x}, \mathbf{y}) = \begin{bmatrix}\nabla_{\mathbf{x}} f_{i,\epsilon}(\mathbf{x}, \mathbf{y}) & -\nabla_{\mathbf{y}} f_{i,\epsilon}(\mathbf{x}, \mathbf{y})\end{bmatrix}^{\top}.$$

The following lemma establishes a connection between $\epsilon$-suboptimal solution of the convex-concave prefix-sum function $g_i$ and $\epsilon$-saddle points of its strongly-convex-strongly-concave augmented function $g_{i,\epsilon}$.

**Lemma D.2** (Luo et al. (2021)). *Suppose that $f(\mathbf{x}, \mathbf{y})$ is convex-concave, $\mathcal{X}$ and $\mathcal{Y}$ are bounded with diameters $D_{\mathcal{X}}$ and $D_{\mathcal{Y}}$ respectively. Consider the function*

$$f_{\epsilon,\mathbf{x}_0,\mathbf{y}_0} := f(\mathbf{x}, \mathbf{y}) + \frac{\epsilon}{8D_{\mathcal{X}}^2}\left\|\mathbf{x} - \mathbf{x}_0\right\|^2 - \frac{\epsilon}{8D_{\mathcal{Y}}^2}\left\|\mathbf{y} - \mathbf{y}_0\right\|^2.$$

*Then for any $(\hat{\mathbf{x}}, \hat{\mathbf{y}}) \in \mathcal{X} \times \mathcal{Y}$, we have*

$$\max_{\mathbf{y}\in\mathcal{Y}} f(\hat{\mathbf{x}}, \mathbf{y}) - \min_{\mathbf{x}\in\mathcal{X}} f(\mathbf{x}, \hat{\mathbf{y}}) \le \frac{\epsilon}{2} + \max_{\mathbf{y}\in\mathcal{Y}} f_{\epsilon,\mathbf{x}_0,\mathbf{y}_0}(\hat{\mathbf{x}}, \mathbf{y}) - \min_{\mathbf{x}\in\mathcal{X}} f_{\epsilon,\mathbf{x}_0,\mathbf{y}_0}(\mathbf{x}, \hat{\mathbf{y}}). \tag{16}$$

---

**Algorithm 4:** Continual Sparse Learning Method (CSL)

**1 Inputs:** $\hat{\mathbf{z}}_0 = (\hat{\mathbf{x}}_0, \hat{\mathbf{y}}_0) \in \mathcal{X} \times \mathcal{Y}$, prev $\leftarrow 0$, flag $\leftarrow$ false, a sequence $\{T_i\}_{i=2}^n$

**2** $\hat{\mathbf{z}}_1 \leftarrow \text{ExtraGradient}(\hat{\mathbf{z}}_0)$, $\tilde{\nabla}_1 \leftarrow h_1(\hat{\mathbf{z}}_1)$

**3 for** $i = 2, \ldots, n$ **do**

**4**     **if** $i - \text{prev} \geq \alpha \cdot i$ **then**

**5**         $\tilde{\mathbf{z}} \leftarrow \hat{\mathbf{z}}_{i-1}$

**6**         $\tilde{\nabla}_{i-1} \leftarrow \frac{1}{i-1} \sum_{j=1}^{i-1} h_{j,\epsilon}(\tilde{\mathbf{z}})$

**7**         prev $\leftarrow i - 1$

**8**         flag $\leftarrow$ true

**9**     **end**

**10**     **Option I:** $\hat{\mathbf{z}}_i \leftarrow \text{SVRG}(\hat{\mathbf{z}}_{\text{prev}}, \tilde{\nabla}_{i-1}, T_i, \{\gamma_t\}, \hat{\mathbf{z}}_{i-1})$

**11**     **Option II:** $\hat{\mathbf{z}}_i \leftarrow \text{SVRE}(\hat{\mathbf{z}}_{\text{prev}}, \tilde{\nabla}_{i-1}, T_i, \{\gamma_t\}, \hat{\mathbf{z}}_{i-1})$

**12**     Select $\hat{u} \sim \text{Unif}(1, \ldots, i-1)$

**13**     $\hat{\nabla}_i \leftarrow \left(1 - \frac{1}{i}\right) \left(h_{\hat{u},\epsilon}(\hat{\mathbf{z}}_i) - h_{\hat{u},\epsilon}(\hat{\mathbf{z}}_{\text{prev}}) + \tilde{\nabla}_{i-1}\right) + \frac{1}{i} h_{i,\epsilon}(\hat{\mathbf{z}}_i)$

**14**     $\tilde{\mathbf{z}}_i \leftarrow \Pi_{\mathcal{X} \times \mathcal{Y}}(\hat{\mathbf{z}}_i - \tau \hat{\nabla}_i)$

**15**     **if** flag **then**

**16**         $\tilde{\mathbf{z}} \leftarrow \hat{\mathbf{z}}_i$

**17**         $\tilde{\nabla}_i \leftarrow \frac{1}{i} \sum_{j=1}^{i} h_{j,\epsilon}(\tilde{\mathbf{z}})$

**18**         prev $\leftarrow i$

**19**         flag $\leftarrow$ false

**20**     **end**

**21**     **else**

**22**         $\tilde{\nabla}_i \leftarrow (1 - \frac{1}{i}) \tilde{\nabla}_{i-1} + \frac{1}{i} h_{i,\epsilon}(\hat{\mathbf{z}}_{\text{prev}})$

**23**     **end**

**24 end**

**25 return:** $\mathbf{x}_T$.

---

Lemma D.2 states that any $\epsilon/2$-suboptimal solution of $g_{i,\epsilon}$ is an $\epsilon$-suboptimal solution of $g_i$. Note that the convergence criteria of $g_{i,\epsilon}$ in the formula (16) is based on the duality gap, while the convergence criteria in Theorem 5.3 and 5.5 are the weighted square of Euclidean distance between the output and the optimum. To establish the convergence result with respect to the duality gap, we incorporate an additional projection iteration on the output $\hat{\mathbf{z}}_i = (\hat{\mathbf{x}}_i, \hat{\mathbf{y}}_i)$ at each stage $i$ in Algorithm 1. In particular, we define an auxiliary variable $\tilde{\mathbf{z}}_i = (\tilde{\mathbf{x}}_i, \tilde{\mathbf{y}}_i)$ with the following update

$$\tilde{\mathbf{z}}_i = \Pi_{\mathcal{X} \times \mathcal{Y}}(\hat{\mathbf{z}}_i - \tau \hat{\nabla}_i),$$

where the stochastic gradient estimator $\hat{\nabla}_i$ is defined as

$$\hat{\nabla}_i = \left(1 - \frac{1}{i}\right) \left(h_{\hat{u},\epsilon}(\hat{\mathbf{z}}_i) - h_{\hat{u},\epsilon}(\hat{\mathbf{z}}_{\text{prev}}) + \tilde{\nabla}_{i-1}\right) + \frac{1}{i} h_{i,\epsilon}(\hat{\mathbf{z}}_i),$$

and $\hat{u}$ is a random index uniformly sampled from $[i-1]$. We can show that the duality gap $\mathbb{E}[\max_{\mathbf{y} \in \mathcal{Y}} g_{i,\epsilon}(\tilde{\mathbf{x}}_i, \mathbf{y}) - \min_{\mathbf{x} \in \mathcal{X}} g_{i,\epsilon}(\mathbf{x}, \tilde{\mathbf{y}}_i)] \leq \epsilon/2$ under appropriate choice of hyperparameters. We present the whole procedure in Algorithm 4.

### D.1 SUPPORTING LEMMAS

We present the convergence analysis of Algorithm 4 with respect to the duality gap during a single stage with the following Lemma.

**Lemma D.3.** *Under Assumption 3.6 and Assumption 3.7 with $\mu > 0$, suppose that a sequence of points $\hat{\mathbf{z}}_i = (\hat{\mathbf{x}}_i, \hat{\mathbf{y}}_i)$ satisfies*

$$\|\hat{\mathbf{x}}_i - \mathbf{x}_i^*\|^2 + \|\hat{\mathbf{y}}_i - \mathbf{y}_i^*\|^2 \leq \epsilon,$$

*where the point $(\mathbf{x}_i^*, \mathbf{y}_i^*)$ is the optimal solution of the minimax optimization problem $\min_{\mathbf{x} \in \mathcal{X}} \max_{\mathbf{y} \in \mathcal{Y}} g_{i,\epsilon}(\mathbf{x}, \mathbf{y})$ for every $i \in [n]$. We introduce*

$$\tilde{\mathbf{z}}_i = \mathcal{P}_{\mathcal{X} \times \mathcal{Y}}(\hat{\mathbf{z}}_i - \tau \hat{\nabla}_i), \tag{17}$$

*where the gradient estimator $\hat{\nabla}_i$ is defined as*

$$\hat{\nabla}_i = \left(1 - \frac{1}{i}\right)\left(h_{\hat{u},\epsilon}(\hat{\mathbf{z}}_i) - h_{\hat{u},\epsilon}(\hat{\mathbf{z}}_{\text{prev}}) + \tilde{\nabla}_{i-1}\right) + \frac{1}{i} h_i(\hat{\mathbf{z}}_i, \epsilon), \quad \hat{u} \sim \text{Unif}(1, \ldots, \mathrm{i} - 1),$$

*then it holds that for every $i \in [n]$,*

$$\mathbb{E}\left[\max_{y \in \mathcal{Y}} g_{i,\epsilon}(\tilde{\mathbf{x}}_i, \mathbf{y}) - g_{i,\epsilon}(\mathbf{x}_i^*, \mathbf{y}_i^*)\right] \leq \left(\sqrt{2}(1 + 5\tau L) + 4(1 + 5\tau L)^2 + 2\right)\varkappa L\epsilon + \frac{\epsilon}{2\tau}$$

$$+ \frac{8GL^3\alpha\tau}{\mu^2}\sqrt{\epsilon} + \frac{128G^2L^4\alpha^2\tau^2}{\mu^3}$$

*and*

$$\mathbb{E}\left[g_{i,\epsilon}(\mathbf{x}_i^*, \mathbf{y}_i^*) - \min_{x \in \mathcal{X}} g_{i,\epsilon}(\mathbf{x}, \tilde{\mathbf{y}}_i)\right] \leq \left(\sqrt{2}(1 + 5\tau L) + 4(1 + 5\tau L)^2 + 2\right)\varkappa L\epsilon + \frac{\epsilon}{2\tau}$$

$$+ \frac{8GL^3\alpha\tau}{\mu^2}\sqrt{\epsilon} + \frac{128G^2L^4\alpha^2\tau^2}{\mu^3}.$$

*Proof.* Let $\tilde{\mathbf{z}}_i = (\tilde{\mathbf{x}}_i, \tilde{\mathbf{y}}_i)$, $\hat{\nabla}_i = (\hat{\nabla}_x^i, -\hat{\nabla}_y^i)$, then the update of $\tilde{\mathbf{z}}_i$ implies that

$$\tilde{\mathbf{x}}_i = \arg\min_{\mathbf{x} \in \mathcal{X}}\left(\langle \hat{\nabla}_{\mathbf{x}}^i, \mathbf{x} - \hat{\mathbf{x}}_i\rangle + \frac{1}{2\tau}\|\mathbf{x} - \hat{\mathbf{x}}_i\|^2\right).$$

Therefore, for any $\mathbf{x} \in \mathcal{X}$, we have

$$\langle \hat{\nabla}_{\mathbf{x}}^i, \mathbf{x} - \hat{\mathbf{x}}_i\rangle + \frac{1}{2\tau}\|\mathbf{x} - \hat{\mathbf{x}}_i\|^2 \geq \langle \hat{\nabla}_{\mathbf{x}}^i, \tilde{\mathbf{x}}_i - \hat{\mathbf{x}}_i\rangle + \frac{1}{2\tau}\|\tilde{\mathbf{x}}_i - \hat{\mathbf{x}}_i\|^2,$$

Rearrange the terms, we have

$$\langle \hat{\nabla}_{\mathbf{x}}^i, \mathbf{x} - \tilde{\mathbf{x}}_i\rangle \geq \frac{1}{2\tau}\left(\|\tilde{\mathbf{x}}_i - \hat{\mathbf{x}}_i\|^2 - \|\mathbf{x} - \hat{\mathbf{x}}_i\|^2\right) \geq -\frac{1}{2\tau}\|\mathbf{x} - \hat{\mathbf{x}}_i\|^2.$$

Taking expectation on both sides and denote $\nabla_{\mathbf{x}} g_{i,\epsilon}(\mathbf{z}) = \frac{1}{i}\sum_{j=1}^i \nabla_{\mathbf{x}} f_{j,\epsilon}(\mathbf{z})$, we have

$$\mathbb{E}[\langle \hat{\nabla}_{\mathbf{x}}^i, \mathbf{x} - \tilde{\mathbf{x}}_i\rangle] = \mathbb{E}[\langle \nabla_{\mathbf{x}} g_{i,\epsilon}(\hat{\mathbf{z}}_i), \mathbf{x} - \tilde{\mathbf{x}}_i\rangle] \geq -\frac{1}{2\tau}\mathbb{E}\left[\|\mathbf{x} - \hat{\mathbf{x}}_i\|^2\right]. \tag{18}$$

Let $\Phi_{g_{i,\epsilon}}(\mathbf{x}) = \max_{\mathbf{y} \in \mathcal{Y}} g_{i,\epsilon}(\mathbf{x}, \mathbf{y})$ and $\mathbf{y}_{g_{i,\epsilon}}^*(\mathbf{x}) = \arg\max_{\mathbf{y} \in \mathcal{Y}} g_{i,\epsilon}(\mathbf{x}, \mathbf{y})$, then it holds that

$$\mathbb{E}[\Phi_{g_{i,\epsilon}}(\tilde{\mathbf{x}}_i) - \Phi_{g_{i,\epsilon}}(\mathbf{x}_i^*)]$$

$$= \mathbb{E}[\Phi_{g_{i,\epsilon}}(\tilde{\mathbf{x}}_i) - \Phi_{g_{i,\epsilon}}(\hat{\mathbf{x}}_i)] - \mathbb{E}[\Phi_{g_{i,\epsilon}}(\mathbf{x}_i^*) - \Phi_{g_{i,\epsilon}}(\hat{\mathbf{x}}_i)]$$

$$\leq \mathbb{E}[\langle \nabla \Phi_{g_{i,\epsilon}}(\hat{\mathbf{x}}_i), \tilde{\mathbf{x}}_i - \hat{\mathbf{x}}_i\rangle] + \varkappa L\|\tilde{\mathbf{x}}_i - \hat{\mathbf{x}}_i\|^2 - \mathbb{E}[\langle \nabla \Phi_{g_{i,\epsilon}}(\hat{\mathbf{x}}_i), \mathbf{x}_i^* - \hat{\mathbf{x}}_i\rangle]$$

$$= \mathbb{E}[\langle \nabla_{\mathbf{x}} g_{i,\epsilon}(\hat{\mathbf{x}}_i, \mathbf{y}_{g_{i,\epsilon}}^*(\hat{\mathbf{x}}_i)), \tilde{\mathbf{x}}_i - \mathbf{x}_i^*\rangle] + \varkappa L\|\tilde{\mathbf{x}}_i - \hat{\mathbf{x}}_i\|^2$$

$$= \mathbb{E}[\langle \nabla_x g_{i,\epsilon}(\hat{\mathbf{x}}_i, \mathbf{y}_{g_{i,\epsilon}}^*(\hat{\mathbf{x}}_i)) - \nabla_{\mathbf{x}} g_{i,\epsilon}(\hat{\mathbf{x}}_i, \hat{\mathbf{y}}_i), \tilde{\mathbf{x}}_i - \mathbf{x}_i^*\rangle] + \mathbb{E}[\langle \nabla_{\mathbf{x}} g_{i,\epsilon}(\hat{\mathbf{x}}_i, \hat{\mathbf{y}}_i), \tilde{\mathbf{x}}_i - \mathbf{x}_i^*\rangle] + \varkappa L\|\tilde{\mathbf{x}}_i - \hat{\mathbf{x}}_i\|^2$$

$$\leq L\mathbb{E}\left[\left\|\mathbf{y}_{g_{i,\epsilon}}^*(\hat{\mathbf{x}}_i) - \hat{\mathbf{y}}_i\right\|\|\tilde{\mathbf{x}}_i - \mathbf{x}_i^*\|\right] + \frac{1}{2\tau}\mathbb{E}[\|\hat{\mathbf{x}}_i - \mathbf{x}_i^*\|^2] + \varkappa L\mathbb{E}[\|\tilde{\mathbf{x}}_i - \hat{\mathbf{x}}_i\|^2],$$

where the first inequality follows from $\Phi_{g_{i,\epsilon}}$ is $2\varkappa L$-smooth and convex by Lemma A.3, the last inequality is due to (18) with $\mathbf{x} = \mathbf{x}^*$ and Cauchy–Schwarz inequality.

According to $\|\mathbf{a} + \mathbf{b}\|^2 \leq 2\|\mathbf{a}\|^2 + 2\|\mathbf{b}\|^2$ and the Lipschitz continuity of $\mathbf{y}_f^*(\cdot)$, we have

$$\mathbb{E}\left[\left\|\hat{\mathbf{y}}_i - \mathbf{y}_{g_{i,\epsilon}}^*(\hat{\mathbf{x}}_i)\right\|^2\right] \leq 2\mathbb{E}\left[\|\hat{\mathbf{y}}_i - \mathbf{y}_i^*\|^2\right] + 2\mathbb{E}\left[\left\|\mathbf{y}_{g_{i,\epsilon}}^*(\hat{\mathbf{x}}_i) - \mathbf{y}_i^*\right\|^2\right]$$

$$\leq 2\mathbb{E}\left[\|\hat{\mathbf{y}}_i - \mathbf{y}_i^*\|^2\right] + 2\varkappa^2\mathbb{E}\left[\|\hat{\mathbf{x}}_i - \mathbf{x}_i^*\|^2\right]$$

$$\leq 2\varkappa^2\epsilon,$$

where the second inequality follows from Lemma A.3 and the last inequality uses the induction hypothesis. Next, the optimality of $\mathbf{x}_i^*$ implies that

$$\mathbf{x}_i^* = \mathcal{P}_{\mathcal{X}}(\mathbf{x}_i^* - \tau\nabla_{\mathbf{x}}g_{i,\epsilon}(\mathbf{x}_i^*, \mathbf{y}_i^*)).$$

Hence, the smoothness of the function $g_{i,\epsilon}$ further implies that

$$\mathbb{E}[\|\tilde{\mathbf{x}}_i - \mathbf{x}_i^*\|] = \mathbb{E}\left[\left\|\mathcal{P}_{\mathcal{X}}(\hat{\mathbf{x}}_i - \tau\hat{\nabla}_{\mathbf{x}}^i) - \mathcal{P}_{\mathcal{X}}(\mathbf{x}_i^* - \tau\nabla_{\mathbf{x}}g_{i,\epsilon}(\mathbf{x}_i^*, \mathbf{y}_i^*))\right\|\right]$$

$$\leq \mathbb{E}\left[\left\|\hat{\mathbf{x}}_i - \mathbf{x}_i^* - \tau(\hat{\nabla}_{\mathbf{x}}^i - \nabla_{\mathbf{x}}g_{i,\epsilon}(\mathbf{x}_i^*, \mathbf{y}_i^*))\right\|\right]$$

$$\leq \mathbb{E}\left[\|\hat{\mathbf{x}}_i - \mathbf{x}_i^* - \tau(\nabla_{\mathbf{x}}g_{i,\epsilon}(\hat{\mathbf{x}}_i, \hat{\mathbf{y}}_i) - \nabla_{\mathbf{x}}g_{i,\epsilon}(\mathbf{x}_i^*, \mathbf{y}_i^*))\|\right] + \tau\mathbb{E}\left[\left\|\hat{\nabla}_{\mathbf{x}}^i - \nabla_{\mathbf{x}}g_{i,\epsilon}(\hat{\mathbf{x}}_i, \hat{\mathbf{y}}_i)\right\|\right]$$

$$\leq \mathbb{E}\left[\|\hat{\mathbf{x}}_i - \mathbf{x}_i^*\|\right] + \tau L\mathbb{E}\left[\sqrt{\|\hat{\mathbf{x}}_i - \mathbf{x}_i^*\|^2 + \|\hat{\mathbf{y}}_i - \mathbf{y}_i^*\|^2}\right] +$$

$$\tau\sqrt{2L^2\mathbb{E}\left[\|\hat{\mathbf{z}}_i - \mathbf{z}_i^*\|^2\right] + \frac{32G^2L^2\alpha^2}{\mu^2} + \frac{4L^2}{\mu}\mathbb{E}\left[\mu\left\|\hat{\mathbf{z}}_{\text{prev}} - \mathbf{z}_{\text{prev}}^*\right\|^2\right]}$$

$$\leq (1 + \tau L)\sqrt{\epsilon} + \sqrt{2}L\tau\sqrt{\epsilon} + \frac{4\sqrt{2}GL\alpha\tau}{\mu} + 2L\tau\sqrt{\epsilon}$$

$$\leq (1 + 5\tau L)\sqrt{\epsilon} + \frac{4\sqrt{2}GL\alpha\tau}{\mu}.$$

The third inequality is due to Assumption 3.7 and Lemma 5.2. Consequently, it follows that

$$\mathbb{E}\left[\max_{\mathbf{y}\in\mathcal{Y}}g_{i,\epsilon}(\tilde{\mathbf{x}}_i, \mathbf{y}) - g_{i,\epsilon}(\mathbf{x}_i^*, \mathbf{y}_i^*)\right]$$

$$\leq \sqrt{2}L\varkappa(1 + 5\tau L)\epsilon + \sqrt{2}L\varkappa\sqrt{\epsilon}\frac{4\sqrt{2}GL\alpha\tau}{\mu} + \frac{\epsilon}{2\tau}$$

$$+ 2\varkappa L\left(\mathbb{E}\left[\|\tilde{\mathbf{x}}_i - \mathbf{x}_i^*\|^2\right] + \left[\|\hat{\mathbf{x}}_i - \mathbf{x}_i^*\|^2\right]\right)$$

$$\leq \sqrt{2}L\varkappa(1 + 5\tau L)\epsilon + \frac{8GL^3\alpha\tau}{\mu^2}\sqrt{\epsilon} + \frac{\epsilon}{2\tau}$$

$$+ 2\varkappa L\left(\epsilon + 2(1 + 5\tau L)^2\epsilon + \frac{64G^2L^2\alpha^2\tau^2}{\mu^2}\right)$$

$$= \left(\sqrt{2}(1 + 5\tau L) + 4(1 + 5\tau L)^2 + 2\right)\varkappa L\epsilon + \frac{\epsilon}{2\tau}$$

$$+ \frac{8GL^3\alpha\tau}{\mu^2}\sqrt{\epsilon} + \frac{128G^2L^4\alpha^2\tau^2}{\mu^3}.$$

Similarly, we can show that

$$\mathbb{E}\left[g_{i,\epsilon}(\mathbf{x}_i^*, \mathbf{y}_i^*) - \min_{\mathbf{x}\in\mathcal{X}}g_{i,\epsilon}(\mathbf{x}, \tilde{\mathbf{y}}_i)\right]$$

$$\leq \left(\sqrt{2}(1 + 5\tau L) + 4(1 + 5\tau L)^2 + 2\right)\varkappa L\epsilon + \frac{\epsilon}{2\tau}$$

$$+ \frac{8GL^3\alpha\tau}{\mu^2}\sqrt{\epsilon} + \frac{128G^2L^4\alpha^2\tau^2}{\mu^3}.$$

$\square$

Under appropriate choice of hyperparameters, we can achieve $\epsilon$-suboptimal solutions at each stage.

**Lemma D.4.** *Following the initial conditions of Lemma D.3, suppose that a sequence of point* $\hat{\mathbf{z}}_i = (\hat{\mathbf{x}}_i, \hat{\mathbf{y}}_i)$ *satisfies that*

$$\mu\|\hat{\mathbf{z}}_i - \mathbf{z}_i^*\|^2 \leq \hat{\epsilon}, \tag{19}$$

where $\mu = \epsilon/(8\max\{D_{\mathcal{X}}^2, D_{\mathcal{Y}}^2\})$ and $\hat{\epsilon} = \mathcal{O}(\epsilon^3/(\max\{G,L\}L))$. We update $\tilde{\mathbf{z}}_i$ by using formula (17) with $\tau = 1/L$ and $\alpha = \mu\hat{\epsilon}^{\frac{1}{3}}G^{-\frac{2}{3}}L^{-\frac{2}{3}}$, then it holds that

$$\mathbb{E}\left[\max_{\mathbf{y}\in\mathcal{Y}} g_{i,\epsilon}(\tilde{\mathbf{x}}_i, \mathbf{y}) - \min_{\mathbf{x}\in\mathcal{X}} g_{i,\epsilon}(\mathbf{x}, \tilde{\mathbf{y}}_i)\right] \leq \frac{\epsilon}{2}.$$

*Proof.* By setting $\tau = 1/L$, $\alpha = \mu\hat{\epsilon}^{\frac{1}{3}}G^{-\frac{2}{3}}L^{-\frac{2}{3}}$ where $\mu = \epsilon/(8\max\{D_{\mathcal{X}}^2, D_{\mathcal{Y}}^2\})$, we have

$$\mathbb{E}\left[\max_{\mathbf{y}\in\mathcal{Y}} g_{i,\epsilon}(\tilde{\mathbf{x}}_i, \mathbf{y}) - \min_{\mathbf{x}\in\mathcal{X}} g_{i,\epsilon}(\mathbf{x}, \tilde{\mathbf{y}}_i)\right]$$

$$\leq 2\left(6\sqrt{2} + 146\right)\frac{L^2\hat{\epsilon}}{\mu^2} + \frac{L\hat{\epsilon}}{\mu} + \frac{16GL^2\alpha}{\mu^{2.5}}\sqrt{\hat{\epsilon}} + \frac{256G^2L^2\alpha^2}{\mu^3}$$

$$\leq \frac{316L^2\hat{\epsilon}}{\mu^2} + \frac{L\hat{\epsilon}}{\mu} + \frac{16G^{\frac{1}{3}}L^{\frac{4}{3}}\hat{\epsilon}^{\frac{5}{6}}}{\mu^{1.5}} + \frac{256G^{\frac{2}{3}}L^{\frac{2}{3}}\hat{\epsilon}^{\frac{2}{3}}}{\mu}$$

$$\leq \frac{317L^2\hat{\epsilon}}{\mu^2} + \frac{16G^{\frac{1}{3}}L^{\frac{4}{3}}\hat{\epsilon}^{\frac{5}{6}}}{\mu^{1.5}} + \frac{256G^{\frac{2}{3}}L^{\frac{2}{3}}\hat{\epsilon}^{\frac{2}{3}}}{\mu}$$

$$\leq \frac{20288L^2\max\{D_{\mathcal{X}}^4, D_{\mathcal{Y}}^4\}\hat{\epsilon}}{\epsilon^2} + \frac{400G^{\frac{1}{3}}L^{\frac{4}{3}}\max\{D_{\mathcal{X}}^3, D_{\mathcal{Y}}^3\}\hat{\epsilon}^{\frac{5}{6}}}{\epsilon^{\frac{3}{2}}}$$

$$+ \frac{2048G^{\frac{2}{3}}L^{\frac{2}{3}}\max\{D_{\mathcal{X}}^2, D_{\mathcal{Y}}^2\}\hat{\epsilon}^{\frac{2}{3}}}{\epsilon}.$$

Therefore, if we choose

$$\hat{\epsilon} = \min\left(\frac{1}{L^2}, \frac{1}{GL}\right) \cdot \frac{\epsilon^3}{262144\max\{D_{\mathcal{X}}^4, D_{\mathcal{Y}}^4\}} = \frac{\epsilon^3}{262144\max\{D_{\mathcal{X}}^4, D_{\mathcal{Y}}^4\}\max\{G,L\}L},$$

then we obtain

$$\mathbb{E}\left[\max_{\mathbf{y}\in\mathcal{Y}} g_{i,\epsilon}(\tilde{\mathbf{x}}_i, \mathbf{y}) - \min_{\mathbf{x}\in\mathcal{X}} g_{i,\epsilon}(\mathbf{x}, \tilde{\mathbf{y}}_i)\right] \leq \frac{\epsilon}{2}.$$

□

Finally, we present the total time complexity of CSVRG and CSVRE methods for solving the continual finite-sum minimax optimization problem in the convex-concave case.

## D.2 PROOF OF COROLLARY 6.2

*Proof.* For the CSVRG method, it takes at most

$$\mathcal{O}\left(\frac{L^2G\log n}{\mu^{\frac{5}{2}}\sqrt{\hat{\epsilon}}} + \frac{nL^2}{\mu^2} + \frac{nG^{\frac{2}{3}}L^{\frac{2}{3}}\log n}{\mu\hat{\epsilon}^{\frac{1}{3}}}\right)$$

$$=\mathcal{O}\left(\frac{L^{\frac{5}{2}}G(G^{\frac{1}{2}} + L^{\frac{1}{2}})\log n}{\epsilon^4} + \frac{nL^2}{\epsilon^2} + \frac{nG^{\frac{2}{3}}L(G^{\frac{1}{3}} + L^{\frac{1}{3}})\log n}{\epsilon^2}\right)$$

$$=\mathcal{O}\left(\frac{L^{\frac{5}{2}}G(G^{\frac{1}{2}} + L^{\frac{1}{2}})\log n}{\epsilon^4} + \frac{nG^{\frac{2}{3}}L(G^{\frac{1}{3}} + L^{\frac{1}{3}})\log n}{\epsilon^2}\right).$$

IFO calls to find a sequence of $\epsilon$-suboptimal solutions.

□

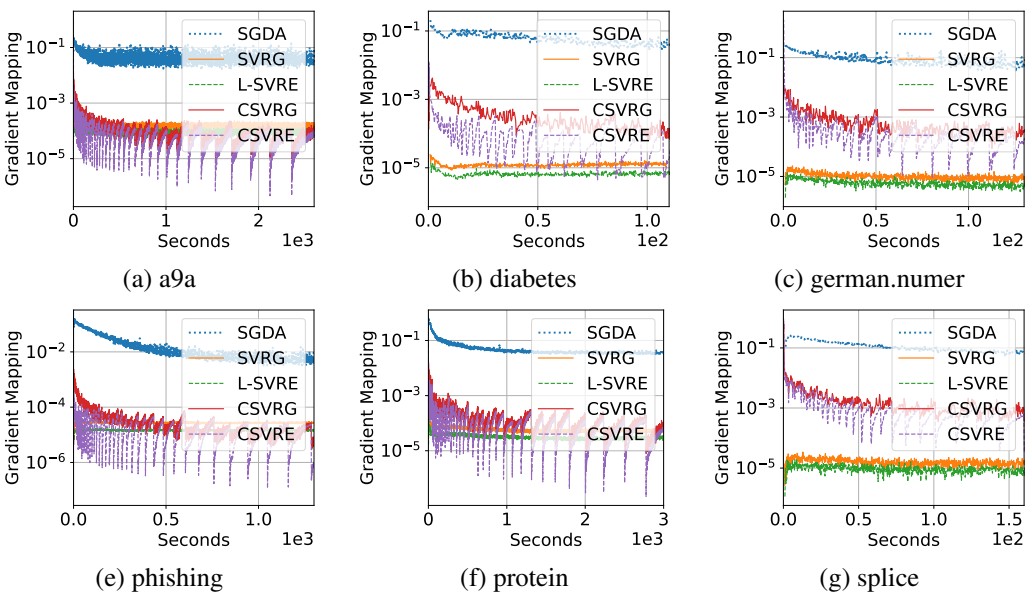

Figure 3: Gradient mapping vs running time for the robust linear regression problem.

### D.3 PROOF OF COROLLARY 6.3

*Proof.* The CSVRE method takes at most

$$\mathcal{O}\left(\frac{LG\log n}{\mu^{\frac{3}{2}}\sqrt{\hat{\epsilon}}} + \frac{L^2 n}{\mu^2} + \frac{G^{\frac{2}{3}}L^{\frac{2}{3}}n\log n}{\mu\hat{\epsilon}^{\frac{1}{3}}}\right)$$

$$=\mathcal{O}\left(\frac{L^{\frac{3}{2}}G(G^{\frac{1}{2}}+L^{\frac{1}{2}})\log n}{\epsilon^3} + \frac{nL^2}{\epsilon^2} + \frac{nG^{\frac{2}{3}}L(G^{\frac{1}{3}}+L^{\frac{1}{3}})\log n}{\epsilon^2}\right)$$

$$=\mathcal{O}\left(\frac{L^{\frac{3}{2}}G(G^{\frac{1}{2}}+L^{\frac{1}{2}})\log n}{\epsilon^3} + \frac{nG^{\frac{2}{3}}L(G^{\frac{1}{3}}+L^{\frac{1}{3}})\log n}{\epsilon^2}\right).$$

IFO calls to find a sequence of $\epsilon$-suboptimal solutions. □

## E ADDITIONAL EXPERIMENTS

In this section, we present additional numerical experiments to demonstrate the effectiveness of our proposed methods.

### E.1 PERFORMANCE VS. RUNNING TIME

In this subsection, we report the empirical results of gradient-mapping versus running time on the robust linear regression problem (Figure 3) and the fairness-aware machine learning problem (Figure 4). The results show that as the running time increases (corresponding to larger datasets), our proposed methods consistently outperform the baseline approaches.

### E.2 PERFORMANCE UNDER BATCHED DATA REVELATION

In this subsection, we modify the experimental setup by revealing ten data points at each iteration instead of one. The empirical results for the robust linear regression problem are presented in Figure 5, and those for the fairness-aware machine learning problem are shown in Figure 6.

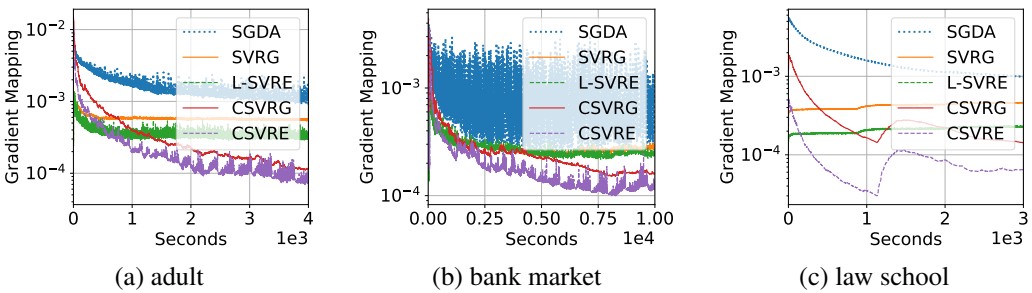

Figure 4: Gradient mapping vs running time for the fairness-aware machine learning problem.

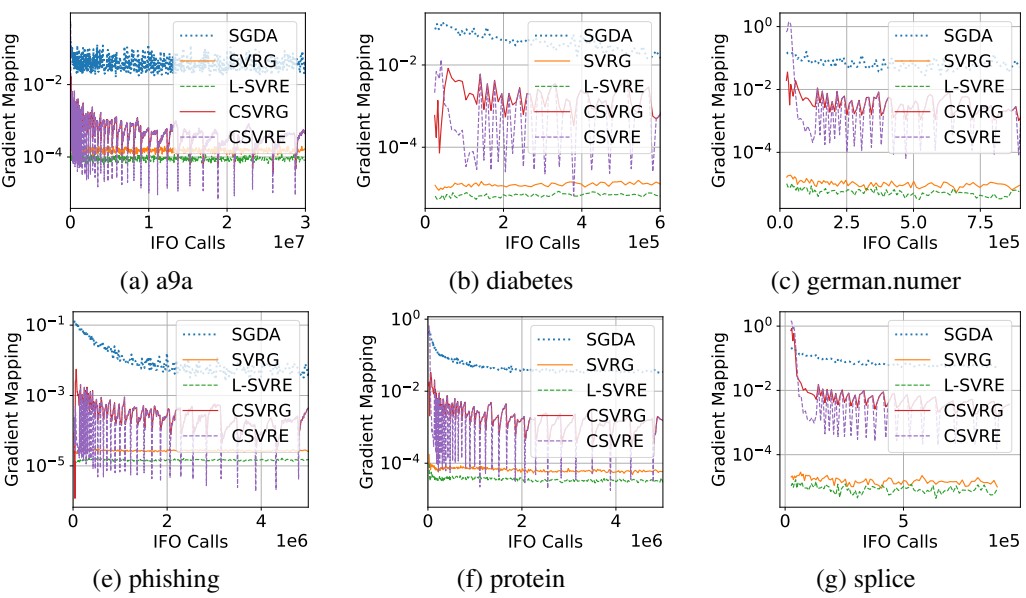

Figure 5: Gradient mapping vs the number of IFO calls for the robust linear regression problem.

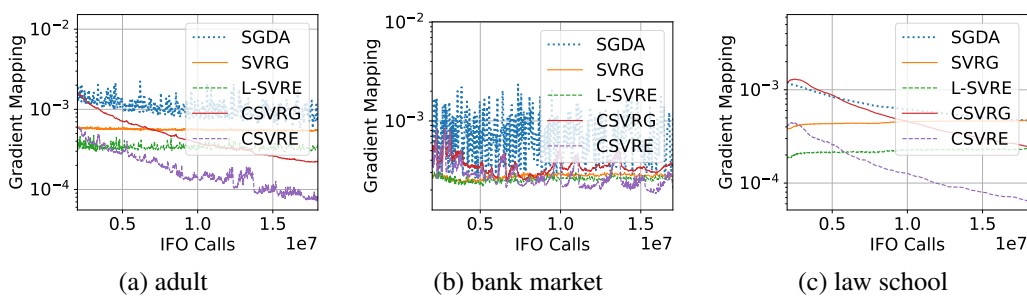

Figure 6: Gradient mapping vs the number of IFO calls for the fairness-aware machine learning problem.

We observe that for the larger problem of fairness-aware machine learning, the total number of iterations remains sufficiently large even when revealing ten points per iteration, and our proposed methods still outperform the existing baselines.

