# OpenReview forum: "Efficient Stochastic Algorithms for Continual Finite-Sum Minimax Optimization"
_ICLR.cc/2026/Conference — Submitted to ICLR 2026_

### Official Review · Reviewer_G6Jx · 2025-10-14

**Soundness:** 3
**Presentation:** 2
**Contribution:** 3
**Rating:** 4
**Confidence:** 3

**Summary:**

The paper analyzes algorithms in the setting of continual finite sum minimax optimization. It analyzes the CSVRG algorithm and proposes a novel extension CSVRE that apart from a constant improvement in the strongly convex strongly concave setting, achieves a rate improvement in the convex concave setting. The paper also provides experimental evaluations for robust linear regression to substantiate its claims.

**Strengths:**

The papers main contributions are the analysis of the CSVRG algorithm for minimax optimization and the proposition and analysis of the CSVRE algorithm for the same setting.

The analysis of both algorithms is interesting and an addition to the minimax literature.

The proposition of the novel extra gradient step to alleviate the strong convexity-concavity constant is interesting and even more so the rate improvement it achieves for the convex-concave setting.

The experimental evaluations corroborate the theoretical findings. They are meaningful extensive and all experiments are under the assumptions and settings over which the authors provide theoretical guarantees.

**Weaknesses:**

The paper has no weakness with respect to its results. The main weakness in my opinion is the writing. The paper presents CSVRG mostly as their own algorithm, while the algorithm was originally presented in [1]. The second algorithm CSVRE is in fact a completely novel contribution therefore I believe that the writing should highlight this difference. In similar cases like in [2], where the authors extended methods from minimization to minimax the algorithms were attributed to the original work.


[1] Efficient Continual Finite-Sum Minimization, Ioannis Mavrothalassitis, Stratis Skoulakis, Leello Tadesse Dadi, Volkan Cevher

[2] Frank-Wolfe Algorithms for Saddle Point Problems, Gauthier Gidel, Tony Jebara, Simon Lacoste-Julien

**Questions:**

My primary question is whether the authors believe the full gradient computation can be fully removed with a variance reduction technique such as STORM [3].

[3] Momentum-Based Variance Reduction in Non-Convex SGD, Ashok Cutkosky, Francesco Orabona

---

> ### Author Response · Authors · 2025-11-18
>
> Thank you for your positive comments and appreciation of our work.
>
> **Q1** The paper presents CSVRG mostly as their own algorithm, while the algorithm was originally presented in [1]. The second algorithm CSVRE is in fact a completely novel contribution therefore I believe that the writing should highlight this difference.
>
> **A1**:
> Thanks for your suggestion.
> In our rebuttal revision (Section 4.1), we have emphasized that our CSVRG is an extension of the algorithm presented in [1].
>
> We would like to clarify that our CSVRG (Section 4.1) is not identical to the algorithm in [1], since our method iterates with the operator
> $$
> \begin{pmatrix}
> \nabla_x f_i(x,y) \\\\
> -\nabla_y f_i(x,y)
> \end{pmatrix}
> $$
> (its stochastic estimate), where the term $-\nabla_y f_i(x,y)$ follows the assumption that $f_i(x,y)$ is concave in $y$ for our minimax optimization.
> In contrast, the algorithm in [1] iterates with the (stochastic) gradient of the component function, since they only consider the convex minimization.
>
> **Q2** My primary question is whether the authors believe the full gradient computation can be fully removed with a variance reduction technique such as STORM [3].
>
> **A2**:
> Thanks for your insightful question.
> Removing the full gradient computation in continual finite-sum minimax optimization is indeed an interesting future direction,
> while how to address it by the algorithm like STORM [3] is still an open problem.
> It is worth noting that our problem setting fundamentally differs from that of STORM [3].
> Specifically, we study the continual finite-sum minimax optimization problem where each component function is strongly-convex–strongly-concave, whereas STORM addresses the stochastic nonconvex minimization problem (no finite-sum structure).
>
> **References**
>
> - [3] Ashok Cutkosky, Francesco Orabona. Momentum-Based Variance Reduction in Non-Convex SGD. NeurIPS 2019.

---

> > ### Comment · Reviewer_G6Jx · 2025-11-24
> >
> > I would like to thank the authors for their reply. My concerns have been alleviated and I have adjusted my score accordingly.

---

> > > ### Author Response · Authors · 2025-11-24
> > >
> > > We are glad to hear that our rebuttal helps address your questions. Thanks for raising our score!

---

### Official Review · Reviewer_yBWa · 2025-10-16

**Soundness:** 3
**Presentation:** 3
**Contribution:** 2
**Rating:** 4
**Confidence:** 4

**Summary:**

This work studies the continual incremental finite sum problem in min-max setting. It extends the technique in [Mavrothalassitis 2024 ] to min-max problems.

**Strengths:**

This paper is technically sound and easy to follow. The discussion on related work is thorough and sufficient for understanding the contribution of this work.

**Weaknesses:**

A first impression can potentially be that applying continual finite sum algorithm from  [Mavrothalassitis 2024 ] to another finite sum VR algorithm by [Alacaoglu 2022] can lack novelty. More discussions on technical challenge can help understanding the contribution.

My major concern for the paper, however, are about the rates:

1. We know that EG is faster than GD for min-max problems, why analyze SVRG whose rate is dominated by SVRE?
2. We know for strongly monotone problems, EG achieve exponential convergence (log(1/ep)), and SVRE in [alacaoglu 2022] achieve exponential convergence + better n dependence. Why can't the proposed method achieve the same dependence in epsilon? Would that cause worse dependence in n / kappa?
3. How good are the proposed method? Can similar lower bounds as in [Mavrothalassitis 2024] be provided?

I think the setup studied in this work is valid, and the analysis is not necessarily simple. However, it looks like that either the rates are hugely suboptimal, or the work lacks enough discussion on dependence tradeoffs. Further, it is hard to identify techinical challenges in the main text.

I can update my score if the authors could address my questions.

Minor:
1.line 239: return xT
2. Would be nice to introduce continual learning optimization setup (e.g., how is suboptimal epsilon defined)
3. The "continual finite sum" is not very well aligned with "continual training / learning" problems in practice, because the model should not be able to see new samples when outputing early weights. It seems a more aligned model would be online decisions / streaming algorithms.

**Questions:**

See weakness

---

> ### Author Response · Authors · 2025-11-18
>
> We thank Reviewer yBWa for your careful review.
>
> **Q1** We know that EG is faster than GD for min-max problems, why analyze SVRG whose rate is dominated by SVRE?
>
> **A1** While it is known that EG can outperform GD in the setting of **classical** finite-sum minimax problem,
> it remains unclear whether this advantage still holds in the setting of **continual** finite-sum minimax problem.
> The prior work has only established convergence results for GD and its variants in continual minimization problems, leaving the minimax case unexplored.
> Therefore, our work provides convergence analyses for both GD and EG in the continual finite-sum minimax optimization setting, thereby extending the theoretical understanding beyond the classical scenario.
>
> **Q2** How good are the proposed method? Can similar lower bounds as in [Mavrothalassitis 2024] be provided?
> I think the setup studied in this work is valid, and the analysis is not necessarily simple. However, it looks like that either the rates are hugely suboptimal, or the work lacks enough discussion on dependence tradeoffs. Further, it is hard to identify techinical challenges in the main text.
>
> **A2** It is possible to provide the lower bound for our continual minimax optimization problem by following the construction of Mavrothalassitis et al. (2024).
> We consider the continual finite-sum minimax optimization problem in which the objective function at the $i$th stage takes the form
> $$
> f_i(x,y) = g_i(x) + h_i(y),
> $$
> where $g_i(x)$ is $\mu$-strongly convex and $h_i(y)$ is $\mu$-strongly concave, which leads to minimizing $x$ and maximizing $y$ are independent.
> It is evident that the lower bound complexity of finding an $\epsilon$-stationary point of $f_i(x,y)$ is no smaller than that of finding an $\epsilon$-stationary point of $g_i(x)$.
> Consequently, following the lower bound analysis for the continual finite-sum minimization problem (Theorems 3 and 5 in Mavrothalassitis et al. (2024)), we can achieve the lower bound of $\Omega(n \epsilon^{-1/4})$.
> Recall that the complexity of our CSVRE method is dominated by the factor of $\mathcal{O}(n \epsilon^{-1/3})$, which is close to the above lower bound.
> However, how to fill the gap of $\epsilon^{-1/12}$ remains an open problem.
>
> The main novelty of our work is that we introduce a novel framework that incorporates extragradient into the continual minimax optimization.
> which is different from existing methods for the continual minimization problem that is based on gradient descent.
> The new iteration scheme leads to our analysis being different from existing work, which yields improved incremental first-order oracle (IFO) complexity in Corollary 5.6 compared to that in Corollary 5.4. Please refer to A4 to Reviewer hrn824 for more details.
>
> **Q3** We know for strongly monotone problems, EG achieve exponential convergence (log(1/eps)), and SVRE in [Alacaoglu 2022] achieve exponential convergence + better n dependence. Why can't the proposed method achieve the same dependence in epsilon?
>
> **A3** In the continual learning setting, we focus on scenarios where the number of stages $n$ is large and the target accuracy $\epsilon$ is of moderate scale.
> Recall that directly applying L-SVRE achieves an IFO complexity of $\mathcal{O}(n^2 \log(1 / \epsilon))$.
> On the other hand, Theorem 3 of Mavrothalassitis et al. (2024) implies that for any $\alpha > 0$, there is no IFO method for the continual optimization problem that can achieve the IFO complexity that is better than $\mathcal{O}(n^{2-\alpha} \log(1 / \epsilon))$.
> Hence, the trade-off between the dependency on $n$ and $\epsilon$ can not be avoided.
> To improve the dependence on $n$, our proposed CSVRE method attains an IFO complexity of $\mathcal{O}(n \epsilon^{-1/3})$,
> but the results in Theorem 3 of Mavrothalassitis et al. (2024) mean it is impossible to retain a logarithmic dependence on $1/\epsilon$ while still achieving a linear dependence on $n$.

---

> ### Author Response · Authors · 2025-11-18
>
> **Q4** Minor: 1.line 239: return xT
>
> 2. Would be nice to introduce continual learning optimization setup (e.g., how is suboptimal epsilon defined)
>
> 3. The "continual finite sum" is not very well aligned with "continual training / learning" problems in practice, because the model should not be able to see new samples when outputing early weights. It seems a more aligned model would be online decisions / streaming algorithms.
>
> **A4**: 1. We have corrected it to $z_T$ in our rebuttal revision.
>
> 2. We formally introduce the continual minimax optimization in lines 41-48, and we define the suboptimal solution in Definition 3.4. (Lines 188-190).
> In particular, we aim to solve the following prefix-sum minimax optimization problem
> $$
>  \min _{x \in \mathcal{X}} \max _{y \in \mathcal{Y}} g _i(x,y) := \frac{1}{i} \sum _{j=1}^i f _j(x,y)
> $$
> at each stage $i \in [n]$, where each component function $f _j$ is $\mu$-strongly-convex-strongly-concave, $L$-smooth and $G$-Lipschitz.
> For $\mu$-strongly-convex-strongly-concave function, a point $(\hat{x}, \hat{y}) \in \mathcal{X} \times \mathcal{Y}$ is said to be an $\epsilon$-saddle point if it satisfies that
>  $\mu||\hat{x} -  x ^*||^2 + \mu || \hat{y} - y  ^ *||^2 \leq \epsilon,$
> where $(x ^ *, y^ *)$ is the optimal solution to the problem.
>
> 3. We appreciate the reviewer’s comment, but we believe there may exist some misunderstanding.
> For our method, achieving the approximate solution $\hat{z}_i$ at the $i$th stage only uses the samples from the first $i$ stages (Lines 4–8 of Algorithm 1, Line 4 of Algorithm 2, and Lines 4–6 of Algorithm 3).
> This means that we do not need to access the future samples $f _j(x)$ for $j > i$.
> Hence, our algorithm is consistent with the continual learning setting where each stage only utilizes past and current data. We welcome further discussion on this point.

---

> > ### Comment · Reviewer_yBWa · 2025-11-21
> > **Response**
> >
> > I thank the authors for the response.
> >
> > 1. The author clarified my question on the optimality by providing the lower bound result in [1]. For this reason, I think this discussion should be included in the intro to justify the rates in this work.
> >
> > 2. Upon reading the response, I still think the analysis on GD instead of EG / Optimistic GDA is redundant. There is no reason / motivation for regular GDA beining efficient in the continual learning setup.
> >
> > In the end, I fail to see how the analysis greatly differs from [1], except that the subprocedure is replaced by variance-reduced EG. Therefore, I will keep my original score.
> >
> > Minor:
> > 1. The current definition for epsilon is confusing in the context of continual learning. It should be stated explicitly that the algorithm outputs xi, yi, upon seeing the ith sample to achieve epsilon-duality gap for the prefix sum up to i samples. This confusion leads to difficulty in understanding line 20 in Alg 1.
> >
> > [1]  Efficient continual finite-sum minimization

---

> ### Author Response · Authors · 2025-11-21
>
> We appreciate the reviewer's further response. We clarify your concerns as follows.
>
> **Q5:** The author clarified my question on the optimality by providing the lower bound result in [1]. For this reason, I think this discussion should be included in the intro to justify the rates in this work.
>
> **A5:** Thanks for your suggestion.
> We have involved the comparison between our IFO complexity and the lower bound in the Introduction of our rebuttal revision and highlighted it by blue (lines 92-93).
> We also include the detailed discussion on the lower bound in Section 5 of our rebuttal revision and highlighted it in blue (lines 378-399).
>
> **Q6:** Upon reading the response, I still think the analysis on GD instead of EG / Optimistic GDA is redundant. There is no reason / motivation for regular GDA beining efficient in the continual learning setup.
>
> **A6** We would like to emphasize that the behavior of GD (GDA) in the continual **minimax** optimization is not well understood, and its analysis differs significantly from that of the EG-based method. For this reason, we believe the analysis of GD is NOT redundant.
>
> Our theoretical results in Corollaries 5.4 and 5.6 show that our GD-based method (CSVRG) and EG-based (CSVRE) method share the same dependence on $n$ and $\epsilon$, differing only by a multiplicative factor of the condition number $\kappa$.
> Hence, when the objective is well-conditioned, CSVRG enjoys a comparable convergence rate to  CSVRE, and its iteration scheme does not require introducing the additional variable $z_i^{t+1/2}$.
>
> **Q7:** In the end, I fail to see how the analysis greatly differs from [1], except that the subprocedure is replaced by variance-reduced EG. Therefore, I will keep my original score.
>
> **A7** We would like to highlight that our analysis of CSVRE  substantially differs from that of Mavrothalassitis et al. [1]. Our analysis is mainly based on controlling **the Euclidean distance between the current iterate and the optimal solution**.
> In particular, we establish Lemma C.1 (1121-1126) to show that our CSVRE satisfies
> $$
> \begin{align}
>          \mu || \hat{z} _ i - z _ i^ *||^2
>       \leq   \frac{ 72 L^2  ||z _ i^ 0 - z_ i^ *||^2}{\mu T_ i^ 2}  + \frac{144 L^2 G^2  \alpha^2 }{ \mu^3 T_ i } +  \frac{18 L^2 \epsilon  }{\mu^2 T_ i}.
> \end{align}
> $$
>
> In the proof of our Theorem 5.5 (1273-1303), we apply the induction and Lemma C.1 with
> $\alpha = \frac{ \mu \epsilon^{\frac{1}{3}}}{G^{\frac{2}{3}} L^{\frac{2}{3}}}$ and
> $$
> \begin{align}
>     T_i = \mathcal{O}(\frac{ L G}{\mu^{\frac{3}{2}}  i \epsilon^{\frac{1}{2}}}+ \frac{L^2 }{\mu^2 } + \frac{ G^\frac{2}{3} L^\frac{2}{3} }{\mu  \epsilon^{\frac{1}{3}}})
> \end{align}
> $$
> to obtain
> $$
>     \mu ||\hat{z} _ i - z _ i^*||^ 2 \leq \epsilon
> $$
> holds for all $i$.
>
> In contrast, Mavrothalassitis et al. [1] derive a recursion in terms of **the function-value gap**. Specifically, their Lemma 9 establishes the result
> $$
> \mathbb{E}[g _ i(x _ i^ {t+1}) - g _ i(x _ i^ *)] \leq \mathbb{E} [\frac{9}{16} \gamma_ t || \nabla_ i^ t - \nabla g_ i (x_ i^ t)||^2 + \frac{1 - \mu \gamma_ t}{2 \gamma_ t} || x_ i^ * - x_ i^ t ||^2 - \frac{1}{2\gamma_ t} || x_ i^ * - x_ i^ {t+1}||^2].
> $$
> It is worth noting that the above upper bound of the function value gap can only work for the analysis of the minimization problem, which cannot be extended to minimax optimization.
> Thus, the framework of our complexity analysis is completely different from that of Mavrothalassitis et al. [1].
>
> **Q8:** The current definition for epsilon is confusing in the context of continual learning. It should be stated explicitly that the algorithm outputs $x_i$, $y_i$, upon seeing the $i$th sample to achieve epsilon-duality gap for the prefix sum up to $i$ samples. This confusion leads to difficulty in understanding line 20 in Alg 1.
>
> **A8:** Thanks for your suggestion. We have revised the presentation of Algorithm 1 by moving the  output step to after achieving $\hat z_i$.
> Please refer to line 11 in Alg 1 of our rebuttal revision.
>
> **References**
> - [1] Ioannis Mavrothalassitis, Stratis Skoulakis, Leello Tadesse Dadi, and Volkan Cevher. Efficient continual finite-sum minimization. In The Twelfth International Conference on Learning Representations, 2024.

---

### Official Review · Reviewer_hrn8 · 2025-10-29

**Soundness:** 3
**Presentation:** 3
**Contribution:** 2
**Rating:** 4
**Confidence:** 4

**Summary:**

This papers studies continual minimax  optimization in both strongly-convex-strongly-concave and convex-concave settings. Particularly, a prefix problem is considered, where stage $i$ tries to find a solution to cumulative stages from 1 to i. Variance reduction techniques are utilized to establish convergence analysis. Experiments have been conducted on robust linear regression and fairness-aware machine learning.

**Strengths:**

1) This paper studies minimax optimization for continual learning, which is underexplored as far as the reviewer is aware of.

2) The convergence analysis and results look well-established.

**Weaknesses:**

1) The requirement of full gradients in Algorithm 1, 2 and 3 make them impractical for large-scale problems.

2) The scale of the experiments are generally small with a lot of them showing lower performance than baselines. Larger scale experiments could make the claims better supported. And maybe more data in one stage instead of one data point at a stage.

3) The considered prefix problem needs to revisit all data of previous stages, which makes it less efficient and not consistent with popular frameworks of continual learning.

4) Looks a lot of the technical results are similar to (Mavrothalassitis et al. (2024)), while the latter is not adequately discussed in the submission. For example, Lemma B.1 and others are similar to results of (Mavrothalassitis et al. (2024)). What is the technical connection between this submission to (Mavrothalassitis et al. (2024), and what are the novelty and challenges on top of the that one?

**Questions:**

How are the baseline methods implemented in experiments to fit in the continual learning setting?
Also, see other comments in Weakness section.

---

> ### Author Response · Authors · 2025-11-18
>
> We thank the Reviewer hrn8 for your insightful review.
>
> **Q1** The requirement of full gradients in Algorithm 1, 2 and 3 make them impractical for large-scale problems.
>
> **A1**
> We would like to clarify that the full gradient $\tilde\nabla _{i-1}$ is computed **infrequently across stages (only in line 5 of Algorithm 1 when $i-$prev$\geq\alpha\cdot i$)**.
> We also emphasize that Algorithms 2 and 3 do **not** require computing the full gradient.
> Specifically, the full gradient $\tilde\nabla _{i-1}$ used in Algorithm 2 (line 4) and Algorithm 3 (lines 4 and 6) has already been achieved by line 5 of Algorithm 1.
> Therefore, we can **reuse** the achieved $\tilde\nabla _{i-1}$ to implement Algorithms 2 and 3, making our method practical for large-scale problems.
>
>
> **Q2**: The scale of the experiments are generally small with a lot of them showing lower performance than baselines. Larger scale experiments could make the claims better supported. And maybe more data in one stage instead of one data point at a stage.
>
> **A2**:
> It is worth noting that the IFO complexity of proposed methods has a better dependency on $n$ (rather than $\epsilon$) than that of baselines (see Tables 1 and 2).
> Hence, the result that our method has lower performance on the small problem is consistent with our theoretical claims.
> We have added the empirical result under the setup of revealing ten data points at each iteration in the appendix (refer to Figures 5 and 6 in Section E.2 in our rebuttal revision). We observe that for the larger problem of fairness-aware machine learning, the total number of iterations remains sufficiently large even when revealing ten points per iteration, and our proposed methods still outperform the existing baselines.
>
> **Q3**: The considered prefix problem needs to revisit all data of previous stages, which makes it less efficient and not consistent with popular frameworks of continual learning.
>
> **A3**: The formulation of our prefix problem is **consistent** with the existing framework for continual learning.
> For example, Mavrothalassitis et al. (2024) investigate the continual finite-sum minimization problem, where the goal is to find a sequence of solutions $x ^* _1, \ldots, x ^* _n$ with each $x ^* _i$ being the solution of the prefix minimization problem
> $$
>     \min _{x \in \mathcal{X}} g _i(x) = \frac{1}{i}\sum _{j=1}^i  f_j(x),
> $$
> where $\mathcal{X}$ is the feasible set.
> It is worth noting that each $g _i$ in the above formulation inherently involves all data from the previous stages.
> We extend Mavrothalassitis et al.'s (2024) framework to the continual finite-sum minimax optimization setting.  In particular, we aim to obtain a sequence of saddle points $(x ^* _1, y ^* _1), \ldots, (x  ^* _n, y ^* _n)$, where each $(x ^* _i, y ^*  _i)$ solves the prefix minimax problem
> $$
>  \min _{x \in \mathcal{X}} \max _{y \in \mathcal{Y}} g _i(x, y) = \frac{1}{i}\sum _{j=1}^i f _j(x, y),
> $$
> where $\mathcal{X}$ and $\mathcal{Y}$ are the feasible sets. Hence, our formulation is a natural and principled extension of the existing continual learning paradigm to the minimax setting.

---

> ### Author Response · Authors · 2025-11-18
>
> **Q4** Looks a lot of the technical results are similar to (Mavrothalassitis et al. (2024)), while the latter is not adequately discussed in the submission. For example, Lemma B.1 and others are similar to results of (Mavrothalassitis et al. (2024)). What is the technical connection between this submission to (Mavrothalassitis et al. (2024), and what are the novelty and challenges on top of the that one?
>
> **A4**  Our work focuses on the minimax problem, which is very different from the minimization problem studied by Mavrothalassitis et al. (2024). In our rebuttal version, we have emphasized that our CSVRG is an extension of Mavrothalassitis et al. (2024) in Section 4.1. We highlight the key differences between the two settings as follows:
>
> First, the convergence analysis of Theorem 5.3 and Corollary 5.4 is mainly based on the monotonicity of the gradient operator (Lemma A.4), while the theory in the minimization problem (Mavrothalassitis et al. 2024) needs the convexity of the objective function, which does not hold in our minimax problem.
>
> Second, we introduce a novel extragradient iteration (as mentioned in Section 4.2) into the continual learning framework, which yields improved incremental first-order oracle (IFO) complexity in Corollary 5.6 compared to that in Corollary 5.4.
> The proposed extragradient iteration can be summarized as follows:
> $$
> \begin{cases}
> \nabla _i^t=\left(1 - \frac{1}{i}\right)\left(h _{u _{t}}(z _i^t) - h _{u _{t,j}}(\hat{z} _{\rm prev})+\tilde{\nabla} _{i-1}\right) + \frac{1}{i}h _i (z _i^t), \\\\
>  z _{i}^{t+1/2} = \Pi _{\mathcal{X} \times \mathcal{Y}}(\bar{z} _i^t - \gamma _t((1 - \frac{1}{i})\tilde{\nabla} _{i-1} + \frac{1}{i} h _i(\hat{z} _{\rm prev}))), \\\\
> z _i^{t+1} \gets \Pi _{\mathcal{X} \times \mathcal{Y}} (\bar{z} _i^t - \gamma _t \nabla _i^{t}).
> \end{cases}
> $$
> Compared with the algorithm for the minimization problem (Mavrothalassitis et al. 2024), our update introduces an additional variable $z_i^{t+1/2}$ to solve the minimax problem.
> Therefore, our main algorithm is completely different from existing methods.
>
> Q5 How are the baseline methods implemented in experiments to fit in the continual learning setting?
>
> A5: We implement the SVRG and L-SVRE in a stagewise manner. At each stage, we choose the outer iteration to be 10 and the inner iterations to be 200. For both baseline methods, the step size is tuned from the set $\\\{1e-5, 3e-5, \dots, 0,1, 0.3\\\}$.
> We highlight more details of the experiment in Section 7 of our revision rebuttal.
>
> **References**
>
> - [Mavrothalassitis et al. 2024] Mavrothalassitis, I., Skoulakis, S., Dadi, L. T., \& Cevher, V. Efficient Continual Finite-Sum Minimization. In The Twelfth International Conference on Learning Representations.

---

### Official Review · Reviewer_Uxcr · 2025-10-31

**Soundness:** 3
**Presentation:** 3
**Contribution:** 3
**Rating:** 6
**Confidence:** 2

**Summary:**

This paper tackles continual finite-sum minimax optimization, where one must solve a sequence of prefix-sum saddle-point problems efficiently. The authors propose two variance-reduced algorithms—CSVRG and CSVRE—that reuse historical gradient information instead of re-solving each stage from scratch. By incrementally updating the full prefix gradient and sparsely refreshing snapshots, they achieve improved IFO complexities in the strongly-convex–strongly-concave case. CSVRE further refines the leading term via an extragradient mid-point.

**Strengths:**

1. A new problem setting that integrates continual learning with minimax optimization, addressing the challenge of efficiently updating solutions across sequential tasks.
2. In particular, the proposed approach sparsely reconstructs the full gradient across stages, significantly reducing redundant computations.
3. It further employs an extragradient scheme, leading to a tighter complexity in terms of the number of IFO.

**Weaknesses:**

1. The presentation could be improved. For example, consider moving the main results for the convex–concave setting into the main text and omitting some of the immediate corollaries to enhance readability and focus.
2. Regarding the experimental results, it would be more informative to report performance curves with respect to running time rather than IFO, to better demonstrate the method’s practical effectiveness.

**Questions:**

1. I am a bit confused by Table 1. It seems there is no explicit $\epsilon$-dependence in the reported results. Could you clarify what convergence metric those baselines use and how it differs from the one adopted in your analysis?
2. The listed baselines are all designed for stochastic or finite-sum settings. Are there any existing baselines tailored to the continual learning setting?

---

> ### Author Response · Authors · 2025-11-18
>
> We thank Reviewer Uxcr for your careful review.
>
> **Q1** The presentation could be improved. For example, consider moving the main results for the convex–concave setting into the main text and omitting some of the immediate corollaries to enhance readability and focus.
>
> **A1** Thanks for your suggestion. We have moved the main results for the convex–concave setting into the main text (refer to Section 6 in our rebuttal revision).
>
> **Q2** Regarding the experimental results, it would be more informative to report performance curves with respect to running time rather than IFO, to better demonstrate the method’s practical effectiveness.
>
> **A2**  We have added the empirical results of gradient-mapping versus running time in the appendix (refer to Figures 3 and 4 in Section E.1 in our rebuttal revision). The results show that as the running time increases (corresponding to larger datasets), our proposed methods consistently outperform the baseline approaches.
>
> **Q3** I am a bit confused by Table 1. It seems there is no explicit $\epsilon$-dependence in the reported results. Could you clarify what convergence metric those baselines use and how it differs from the one adopted in your analysis?
>
> **A3** In Table 1, the IFO complexity of all methods contains the dependency on $\log(1/\epsilon)$.
> Since we adopt the notation $\tilde{O}(\cdot)$, this logarithmic factor with respect to $\epsilon$ is omitted.
> Both our methods and all baseline methods use the expected squared Euclidean distance to the optimal solution (i.e., $\mathbb{E}[||z_k - z^*||^2]$) as the convergence metric.
> We explicitly highlight this convergence metric in the caption of Table 1 in our rebuttal revision (line 57).
>
> **Q4** The listed baselines are all designed for stochastic or finite-sum settings. Are there any existing baselines tailored to the continual learning setting?
>
>
> **A4** To the best of our knowledge, there is no existing baseline that is specifically designed for the continual minimax optimization setting. Our work is the first to propose stochastic methods explicitly tailored to the continual learning scenario.

---

### Author Response · Authors · 2025-12-01

We thank the area chairs and reviewers for their careful evaluations and helpful comments. We also appreciate the positive feedback from multiple reviewers, e.g.,

* A new problem setting that integrates continual learning with minimax optimization, addressing the challenge of efficiently updating solutions across sequential tasks.

* The convergence analysis and results look well-established.

* This paper is technically sound and easy to follow. The discussion on related work is thorough and sufficient for understanding the contribution of this work.

* The proposition of the novel extra gradient step to alleviate the strong convexity-concavity constant is interesting and even more so the rate improvement it achieves for the convex-concave setting.

* The experimental evaluations corroborate the theoretical findings. They are meaningful extensive and all experiments are under the assumptions and settings over which the authors provide theoretical guarantees.

We have addressed the reviewers' concerns in rebuttal and revised our submission accordingly (highlight the changes in blue).
Due to the discussion period has been cut short, we have only received the response from Reviewers yBWa and G6Jx.
We also greatly appreciate Reviewer G6Jx for the positive assessment of our work and for raising the score.

We summarize the main points in our rebuttal:

**1. Reviewer Uxcr (Q2) suggested  reporting performance curves with respect to running time rather than IFO**

We have added the empirical results of gradient-mapping versus running time in the appendix (refer to Figures 3 and 4 in Section E.1 in our rebuttal revision). The results show that as the running time increases (corresponding to larger datasets), our proposed methods consistently outperform the baseline approaches.

**2. Reviewer hrn8 (Q1) argued that the requirement of full gradients in Algorithm 1, 2 and 3 is impractical for large-scale problems.**

We would like to clarify that the full gradient $\tilde\nabla _{i-1}$ is computed infrequently across stages (only in line 5 of Algorithm 1 when $i-$prev$\geq\alpha\cdot i$). We also emphasize that Algorithms 2 and 3 do not require computing the full gradient. Specifically, the full gradient $\tilde\nabla _{i-1}$ used in Algorithm 2 (line 4) and Algorithm 3 (lines 4 and 6) has already been achieved by line 5 of Algorithm 1. Therefore, we can reuse the achieved $\tilde\nabla _{i-1}$ to implement Algorithms 2 and 3, making our method practical for large-scale problems.

**3. Reviewer hrn8 (Q4) and  Reviewer yBWa (Q7) appear to be confused about whether the technical results of our work are similar to those of Mavrothalassitis et al. (2024).**

Our work focuses on the minimax problem, which is very different from the minimization problem studied by Mavrothalassitis et al. (2024). We highlight the key differences between the two settings as follows:

First, the convergence analysis of Theorem 5.3 and Corollary 5.4 is mainly based on the monotonicity of the gradient operator (Lemma A.4), while the theory in the minimization problem (Mavrothalassitis et al. 2024) needs the convexity of the objective function, which does not hold in our minimax problem.
Furthermore, our analysis is mainly based on controlling the Euclidean distance between the current iterate and the optimal solution, while Mavrothalassitis et al. (2024) derive a recursion in terms of the function-value gap.

Second, we introduce a novel extragradient iteration (as mentioned in Section 4.2) into the continual learning framework, which yields improved incremental first-order oracle (IFO) complexity in Corollary 5.6 compared to that in Corollary 5.4. The proposed extragradient iteration can be summarized as follows:
 $$
\begin{cases}
\nabla _i^t=\left(1 - \frac{1}{i}\right)\left(h _{u _{t}}(z _i^t) - h _{u _{t,j}}(\hat{z} _{\rm prev})+\tilde{\nabla} _{i-1}\right) + \frac{1}{i}h _i (z _i^t), \\\\
 z _{i}^{t+1/2} = \Pi _{\mathcal{X} \times \mathcal{Y}}(\bar{z} _i^t - \gamma _t((1 - \frac{1}{i})\tilde{\nabla} _{i-1} + \frac{1}{i} h _i(\hat{z} _{\rm prev}))), \\\\
 z _i^{t+1} \gets \Pi _{\mathcal{X} \times \mathcal{Y}} (\bar{z} _i^t - \gamma _t \nabla _i^{t}).
\end{cases}
$$

Compared with the algorithm for the minimization problem (Mavrothalassitis et al. 2024), our update introduces an additional variable $z_i^{t+1/2}$ to solve the minimax problem. Therefore, our main algorithm is completely different from existing methods.

---

### Author Response · Authors · 2025-12-01

**4. Reviewer yBWa (Q2) is confused about the lower bound of our setting.**

It is possible to provide the lower bound for our continual minimax optimization problem by following the construction of Mavrothalassitis et al. (2024). We consider the continual finite-sum minimax optimization problem in which the objective function at the $i$th stage takes the form $$ f_i(x,y) = g_i(x) + h_i(y), $$ where $g_i(x)$ is $\mu$-strongly convex and $h_i(y)$ is $\mu$-strongly concave, which leads to minimizing $x$ and maximizing $y$ are independent. It is evident that the lower bound complexity of finding an $\epsilon$-stationary point of $f_i(x,y)$ is no smaller than that of finding an $\epsilon$-stationary point of $g_i(x)$. Consequently, following the lower bound analysis for the continual finite-sum minimization problem (Theorems 3 and 5 in Mavrothalassitis et al. (2024)), we can achieve the lower bound of $\Omega(n \epsilon^{-1/4})$. Recall that the complexity of our CSVRE method is dominated by the factor of $\mathcal{O}(n \epsilon^{-1/3})$, which is close to the above lower bound. However, how to fill the gap of $\epsilon^{-1/12}$ remains an open problem.

**5. Reviewer G6Jx (Q1) suggested polishing the presentation of CSVRG.**

In our rebuttal revision (Section 4.1), we have emphasized that our CSVRG is an extension of the algorithm presented in (Mavrothalassitis et al. 2024).

---

### Meta-Review · Area_Chair_mwor · 2026-01-04

**Summary:**

This paper considers continual finite-sum convex–concave minimax optimization. It proposes an efficient stochastic first-order algorithm to solve this problem and establishes its theoretical convergence rate. Numerical experiments are provided to demonstrate the performance of the proposed algorithms.

The reviewers acknowledge that the problem setup is novel and appreciate the theoretical results. However, they also raise several concerns regarding the efficiency of the algorithm, the novelty, the scale of the experiments, and inferior performance in certain settings. For example, the first algorithm requires computing full gradients, which limits its practicality in real-world applications. In addition, the novelty is incremental, as the proof largely follows existing work on continual minimization problems. Only small applications are used to validate the performance of the proposed algorithm.

Because those concerns were not adequately addressed in the rebuttal, I recommend rejection.

**Reviewer Concerns:**

The reviewers acknowledge that the problem setup is novel and appreciate the theoretical results. However, they also raise several concerns regarding the efficiency of the algorithm, the novelty, the scale of the experiments, and inferior performance in certain settings. For example, the first algorithm requires computing full gradients, which limits its practicality in real-world applications. In addition, the novelty is incremental, as the proof largely follows existing work on continual minimization problems. Only small applications are used to validate the performance of the proposed algorithm. Those concerns were not adequately addressed in the rebuttal.

**Reviewer Scores:**

Because the concerns were not adequately addressed in the rebuttal, the overall evaluation is still negative.

---

### Decision · Program_Chairs · 2026-01-26

Reject